# Riemannian Metric Matching
# for Scalable Geometric Modeling of Distributions

**Jacob Bamberger** [1]   **Adam Gosztolai** [2]   **Pierre Vandergheynst** [3]   **Michael Bronstein** [1 4]   **Iolo Jones** [1]

## Abstract

High-dimensional datasets often concentrate near low-dimensional structures, but estimating their geometry from samples typically relies on graphs and kernels that scale poorly with dataset size and dimension. We propose Riemannian metric matching: a denoising probabilistic framework for learning the Riemannian geometry of data using neural networks. Specifically, we learn the *carré du champ* operator, which, using diffusion geometry, gives us access to the Riemannian geometry toolkit for downstream machine learning and statistical tasks. Our key observation is that the carré du champ operator can be formulated as a conditional expectation over random perturbations of the data, which can be exploited for sample-wise training and constant cost, amortized inference without explicit kernel construction. Empirically, metric matching rivals or improves the accuracy of $k$-NN-based diffusion geometry estimators, while enabling amortized inference that is up to $400\times$ faster, and supports graph-free geometric analysis on high-dimensional images where nearest neighbors break down.

## 1. Introduction

Although high-dimensional datasets are typical in machine learning, empirical evidence across domains suggests that the local intrinsic structure of many real-world datasets is effectively low-dimensional, commonly referred to as the manifold hypothesis. The existence of this lower dimensional structure has led to the use of geometric or spatial

---

[1]Department of Computer Science, University of Oxford, Oxford, UK [2]Institute of Artificial Intelligence, Medical University of Vienna, Vienna, Austria [3]EPFL, Lausanne, Switzerland [4]AITHYRA. Correspondence to: Jacob Bamberger <jacob.bamberger@cs.ox.ac.uk>, Iolo Jones <iolo.jones@cs.ox.ac.uk>.

*Proceedings of the 43$^{rd}$ International Conference on Machine Learning*, Seoul, South Korea. PMLR 306, 2026. Copyright 2026 by the author(s).

tools on diverse data types. These tools can probe intrinsic properties of data, including dimensionality, tangent spaces, and curvature, to name a few. Such methods have been successfully applied in areas ranging from computer vision (Murase & Nayar, 1993) and generative modeling (Song & Ermon, 2019; Bamberger et al., 2026) to single-cell data (Moon et al., 2019) and molecular dynamics (Facco et al., 2017; Diepeveen et al., 2024). Beyond applications, the manifold hypothesis plays an important theoretical role, offering insight into how deep neural networks generalize in high dimensions and mitigate the curse of dimensionality (Bengio et al., 2013; Pope et al., 2021; Kiani et al., 2024).

The first step in a standard geometric data analysis pipeline is to build a graph whose nodes correspond to data points and whose edges encode local connectivity. The graph is either dense and weighted by a kernel function of the distance between the nodes, or sparse, such as $k$-nearest neighbor ($k$-NN) graphs, possibly unweighted, with downstream analysis often sensitive to the hyper-parameter $k$. While strong convergence guarantees hold in the infinite-data limit (Coifman & Lafon, 2006), these approaches become computationally prohibitive at scale. In particular, both computational and memory costs grow super-linearly, if not quadratically, with dataset size. Crucially, these costs persist at inference time, as processing new points require recomputing nearest neighbors for each query, preventing efficient out-of-sample extension. These graph and kernel methods are also limited in their practical scope by their reliance on the pairwise distances between points, which cease to be discriminatory in high dimensions, and miss out on the many advantages of representation learning with neural networks that have made deep learning so successful (LeCun et al., 2015).

A more recent line of work leverages trained neural networks – such as VAEs, GANs, or diffusion models – to recover geometric information (Stanczuk et al., 2024), often through the Jacobian of the trained network (Arvanitidis et al., 2018). Since these networks are typically trained for representation learning or generative modeling rather than geometric estimation, existing approaches historically provide few theoretical guarantees that the recovered geometric quantities converge to the true underlying geometry in the infinite-data limit. In the diffusion setting, however,

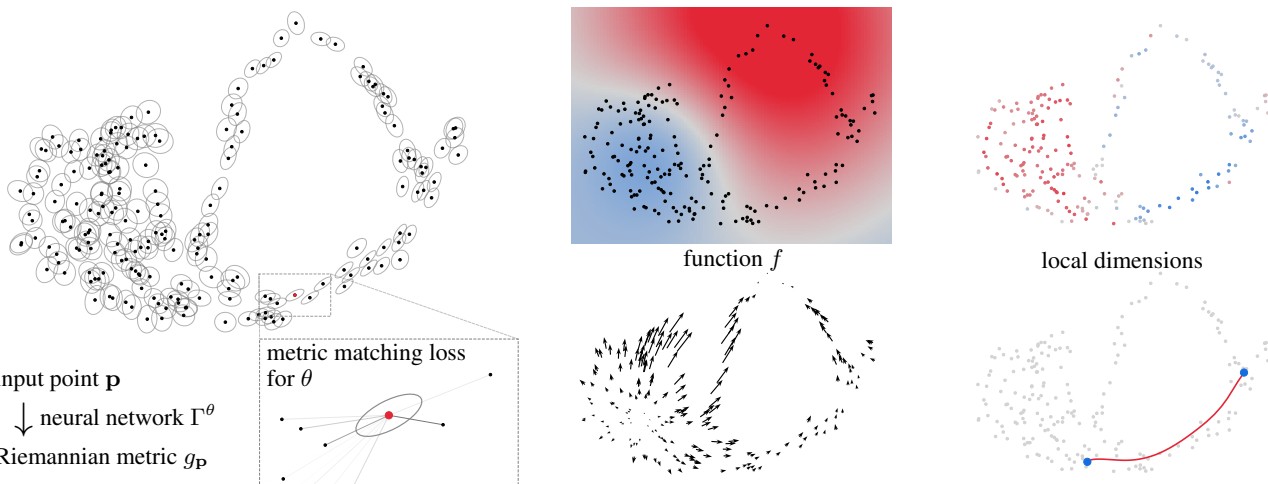

input point **p**

↓ neural network $\Gamma^\theta$

Riemannian metric $g_{\mathbf{p}}$

metric matching loss for $\theta$

function $f$

local dimensions

intrinsic gradient $\nabla f$

interpolating paths

*Figure 1.* **Intrinsic diffusion geometry with Riemannian metric matching.** The neural network learns a Riemannian metric (ellipses, top left) by implicitly averaging multiple rank-1 contributions through the metric matching loss (inset, bottom left). This gives us access to the toolkit of Riemannian geometry methods via diffusion geometry. Given a scalar field $f$ (top middle), we can compute its intrinsic gradient $\nabla f$ with respect to the data geometry (bottom middle). We estimate local data dimensions from the ratio of the metric eigenvalues (top right: red corresponds to higher dimension), and on-manifold interpolating paths between pairs of points (bottom right).

Kharitenko et al. (2025) recently showed that the Jacobian of a Gaussian denoiser converges to the tangent-space projector, and used this for Riemannian optimization on data manifolds. These approaches avoid pairwise distances and $k$-NN graphs, but shift the cost to computing Jacobians of large networks at inference time, which can be expensive and numerically unstable in high dimensions. Additionally, the geometry is extracted indirectly from a pretrained representation or generative model, rather than learned directly.

In this work, we train a neural surrogate to regress the metric directly, rather than extracting geometry from neighborhood graphs or from derivatives of a pretrained models. Our main contribution is a training objective inspired by the denoising objectives from diffusion models (Sohl-Dickstein et al., 2015; Ho et al., 2020; Song et al., 2021). This loss allows a neural network to learn a Riemannian metric at every point by rewriting an intractable quantity as the marginalization of tractable ones. Specifically, we learn an estimate for the *carré du champ* operator (Bakry et al., 2013), which can be used to recover the Riemannian geometry of the data via diffusion geometry (Jones, 2024a). Given finite data, the minimizer of the loss corresponds to classical kernel-based geometric estimates, and in the infinite-data regime, the minimizer recovers the desired geometric quantities when the data distribution is locally supported on a manifold.

We demonstrate our method by applying intrinsic dimensionality estimation and tangent space recovery in high-dimensional settings where graph-based diffusion geometry becomes unreliable. We also apply it to Riemannian optimization, where we compute intrinsic gradients to recover interpolating paths between data points that remain near data manifold. By replacing neighborhood graphs with a learned, denoising-based surrogate, our approach makes diffusion-geometric analysis feasible in high-dimensional and large-scale settings where classical methods break down. This allows the design of novel deep learning methods by directly inheriting tools from Riemannian geometry (Jones & Lanners, 2026).

**Contributions:**

- We propose a neural surrogate for the carré du champ operator that enables amortized, dataset-size–independent inference of local geometric quantities without neighborhood graphs.

- We introduce a denoising-based, sample-wise **conditional Riemannian metric matching** objective that enables scalable training of the surrogate without explicit kernel construction or pairwise distance computations.

- We establish a theoretical connection between the denoising objective and classical diffusion maps, guaranteeing recovery of geometric quantities in the limit when the data is locally sampled from a manifold.

- We demonstrate empirically that metric matching recovers the correct geometry on known data manifolds, and leads to meaningful tangent vectors and high-quality on-manifold interpolating paths along high-dimensional image data.

## 2. Background

**Manifolds and Riemannian metrics.** A $d$-dimensional manifold $\mathcal{M}$ is a topological space that is locally homeomorphic to $\mathbb{R}^d$. A *Riemannian metric* $g$ on $\mathcal{M}$ assigns to each point $p \in \mathcal{M}$ an inner product $g_p(\cdot, \cdot)$ on the tangent space

$T_p\mathcal{M}$, varying smoothly with $p$. The metric induces geometric notions such as lengths of curves, angles, volumes, and the Laplace–Beltrami operator $\Delta_g$.

In the case that $\mathcal{M}$ is a smooth $d$-dimensional submanifold of $\mathbb{R}^D$, the tangent space $T_{\mathbf{p}}\mathcal{M}$ is a $d$-dimensional linear subspace of $\mathbb{R}^D$, and $\mathcal{M}$ inherits a Riemannian metric from the ambient space by restricting the Euclidean inner product $\langle \mathbf{u}, \mathbf{v} \rangle = \mathbf{u}^\top \mathbf{v}$ to vectors in $T_{\mathbf{p}}\mathcal{M}$. This choice of metric makes $\mathcal{M}$ an isometrically embedded submanifold of $\mathbb{R}^D$. If $f : \mathbb{R}^D \to \mathbb{R}$ is smooth, its *intrinsic gradient* $\nabla f$ is a vector field on $\mathcal{M}$. At each $\mathbf{p} \in \mathcal{M}$, $\nabla_{\mathbf{p}} f \in T_p\mathcal{M}$ is a tangent vector pointing in the direction of steepest increase of $f$ along the manifold $\mathcal{M}$, and $\nabla_{\mathbf{p}} f$ coincides with the projection of the ambient Euclidean gradient (Jacobian)

$$\boldsymbol{\partial} f(\mathbf{p}) := \left( \frac{\partial f}{\partial x_1}(\mathbf{p}), ..., \frac{\partial f}{\partial x_D}(\mathbf{p}) \right) \in \mathbb{R}^D \qquad (1)$$

at $\mathbf{p}$ onto the tangent space $T_{\mathbf{p}}\mathcal{M}$.

**Diffusion geometry and the carré du champ.** Classical Riemannian geometry applies on manifolds, but real-world data rarely satisfy the stringent conditions of being a manifold (like having no branching points, and a uniformly constant local dimension). Diffusion geometry (Jones, 2024a) describes the geometry of more general spaces through the behavior of diffusion processes on them. This applies to the low-dimensional but non-manifold data geometries that occur in practice, allowing the transfer of Riemannian geometry methods to generic data geometries.

On a Riemannian manifold $(\mathcal{M}, g)$, the canonical diffusion is Brownian motion $dx = dB_t$, whose evolution is characterized by its *infinitesimal generator*, the Laplace-Beltrami operator $\Delta_g$. The diffusion is related to the Riemannian metric $g$ by the carré du champ (CDC) identity

$$g(\nabla f, \nabla h) = \frac{1}{2} \left( f\Delta_g h + h\Delta_g f - \Delta_g(fh) \right) \qquad (2)$$

where $f, h : \mathcal{M} \to \mathbb{R}$ are two smooth scalar fields. In the theory of Markov diffusion operators (Bakry et al., 2013), the infinitesimal generator $\mathcal{L}$ of any Markov process defines a bilinear form called the *carré du champ operator*

$$\Gamma_{\mathcal{L}}(f, h) := \frac{1}{2}(f\mathcal{L}h + h\mathcal{L}f - \mathcal{L}(fh)) \qquad (3)$$

so that $\Gamma_{\Delta_g}(f, h) = g(\nabla f, \nabla h)$ on a manifold. In diffusion geometry, this relationship is abstracted and used to define a generalized Riemannian geometry relative to an arbitrary Markov diffusion process (Jones, 2024a). Specifically, the carré du champ operator of a general Markov process is used to *define* the Riemannian metric via Eq. 2, from which we can construct the rest of Riemannian geometry. This means that, by learning the carré du champ, we can access the toolkit of classical Riemannian geometry methods, such as intrinsic calculus, curvature, and topology (Jones, 2024a;b).

## 3. Conditional metric matching

In practice, we aim to learn the geometry of an underlying dataset by observing samples $X \sim p$, and estimating the carré du champ operator. When the data lie on a manifold, a standard approach to approximate $\Delta_g$, uses the heat kernel

$$w_\varepsilon (\mathbf{x}, \mathbf{y}) = \exp \left( -\frac{\|\mathbf{x} - \mathbf{y}\|^2}{2\varepsilon} \right)$$

with bandwidth $\varepsilon$. This approach was originally introduced to justify kernel methods, but will also apply to metric matching. We define the normalized diffusion operator

$$(P_\varepsilon f)(\mathbf{y}) := \frac{(K_\varepsilon f)(\mathbf{y})}{d_\varepsilon(\mathbf{y})},$$

where

$$(K_\varepsilon f)(\mathbf{y}) := \mathbb{E}_{X \sim p(x)}[w_\varepsilon(\mathbf{y}, X)f(X)],$$
$$d_\varepsilon(\mathbf{y}) := \mathbb{E}_{X \sim p(x)}[w_\varepsilon(\mathbf{y}, X)]$$

are the kernel operator and a corresponding degree function, respectively. Intuitively, $(P_\varepsilon f)(\mathbf{y})$ is a local average of $f$ near $\mathbf{y}$ under a Gaussian window of scale $\varepsilon$. When $p$ is supported on a manifold $\mathcal{M}$, and

$$\mathcal{L}f := \Delta_g f - 2g(\nabla \log p, \nabla f), \qquad (4)$$

the operator $\mathcal{L}_\epsilon := (\mathbf{I} + P_\varepsilon)/\varepsilon$ converges pointwise to $\mathcal{L}$ as $\varepsilon \to 0$ (Coifman & Lafon, 2006; Belkin & Niyogi, 2003).

A key property of the CDC operator is that it does not depend on the first-order drift terms, which cancel out when Eq. 4 is substituted into Eq. 3 by the Leibniz rule, so $\Gamma_{\mathcal{L}} = \Gamma_{\Delta_g}$ (see Appendix A). This means that the approximation $\Gamma_\epsilon := \Gamma_{\mathcal{L}_\epsilon}$ converges pointwise to the Riemannian metric tensor $\Gamma_{\mathcal{L}} = \Gamma_{\Delta_g}$. We can substitute $\mathcal{L}_\epsilon = (\mathbf{I} - P_\varepsilon)/\varepsilon$ into Eq. 3 to rewrite $\Gamma_\varepsilon(f, h)(\mathbf{y})$ as the expectation

$$\frac{\mathbb{E}_X \left[ w_\varepsilon(\mathbf{y}, X)\big(f(X) - f(\mathbf{y})\big)\big(h(X) - h(\mathbf{y})\big) \right]}{2\varepsilon \, \mathbb{E}_X[w_\varepsilon(\mathbf{y}, X)]}, \qquad (5)$$

which can be interpreted as the *uncentered* local covariance between $f$ and $h$ under the scale-$\varepsilon$ diffusion generated by $P_\varepsilon$. We also consider the *centered* local covariance replacing $f(\mathbf{y})$ and $h(\mathbf{y})$ by $(P_\varepsilon f)(\mathbf{y})$ and $(P_\varepsilon h)(\mathbf{y})$

$$\frac{\mathbb{E}_X \left[ w_\varepsilon(\mathbf{y}, X)\big(f(X) - (P_\varepsilon f)(\mathbf{y})\big)\big(h(X) - (P_\varepsilon h)(\mathbf{y})\big) \right]}{2\varepsilon \, \mathbb{E}_X[w_\varepsilon(\mathbf{y}, X)]}. \qquad (6)$$

These expectations can be estimated from samples using kernel-weighted neighborhood graphs, but this becomes computationally and memory intensive for many samples, and suffers from the curse of dimensionality in high dimensions. In the next section, we show how to construct a scalable loss function and training procedure that allows a neural network to estimate $\Gamma_\varepsilon(f, h)$ from samples.

## 3.1. Conditional carré du champ matching

Our goal is to train a neural network to compute or approximate $\Gamma_\varepsilon$ in Eq. 5. A natural objective is to regress $\Gamma_\varepsilon$ under the smoothed density $p_Y = p * \mathcal{N}(0, \varepsilon\mathbf{I})$:

$$\mathcal{L}_{marg}^{CDC}(\theta) := \mathop{\mathbb{E}}_{Y}\left[\left(\Gamma_\varepsilon^\theta(Y) - \Gamma_\varepsilon(f,h)(Y)\right)^2\right]. \quad (7)$$

However, as we have seen, $\Gamma_\varepsilon(f,h)(\mathbf{y})$ from Eq. 5, which we will refer to as the *marginal* CDC, is intractable. We therefore introduce a tractable *conditional* CDC, defined as

$$\Gamma_\varepsilon(f,h)(\mathbf{x},\mathbf{y}) := \frac{1}{2\varepsilon}\left(f(\mathbf{x}) - f(\mathbf{y})\right)\left(h(\mathbf{x}) - h(\mathbf{y})\right).$$

Since we chose $p_Y = p * \mathcal{N}(0, \varepsilon\mathbf{I})$, we have

$$p_Y(\mathbf{y}|\mathbf{x}) = (2\pi\varepsilon)^{-D/2}w_\varepsilon(\mathbf{y},\mathbf{x})$$

$$d_\varepsilon(\mathbf{y}) = (2\pi\varepsilon)^{D/2}\mathbb{E}_X[p_Y(\mathbf{y}|X)] = (2\pi\varepsilon)^{D/2}p_Y(\mathbf{y}),$$

so we can use Bayes' rule to rewrite the marginal CDC as the following conditional expectation

$$\Gamma_\varepsilon(f,h)(\mathbf{y}) = \frac{\mathbb{E}_X\left[p_Y(\mathbf{y}|X)\Gamma_\varepsilon(f,h)(X,\mathbf{y})\right]}{p_Y(\mathbf{y})} \quad (8)$$

$$= \mathop{\mathbb{E}}_{X}\left[\Gamma_\varepsilon(f,h)(X,\mathbf{y}) \mid Y = \mathbf{y}\right]. \quad (9)$$

We now introduce the tractable conditional loss:

$$\mathcal{L}_{cond}^{CDC}(\theta) := \mathop{\mathbb{E}}_{X,Y|X}\left[\left(\Gamma_\varepsilon^\theta(Y) - \Gamma_\varepsilon(f,h)(X,Y)\right)^2\right] \quad (10)$$

where $X \sim p$ and $Y|X \sim \mathcal{N}(X, \varepsilon\mathbf{I})$. This loss can be sampled from, evaluated, and backpropagated through easily since $X$ is sampled from the data distribution $p$, $Y$ is a noisy version of $X$, and the conditional CDC is computed in $\mathcal{O}(1)$ since it does not involve an expectation or a normalizing constant. The expectation decomposes over independent samples $X$ and $Y|X$, so it trivially parallelizable and well suited for modern GPU-based mini-batch and distributed training. Perhaps surprisingly, the conditional loss is equal to the marginal loss up to a constant independent of the parameters $\theta$, so the gradients of both losses are equal. As such, optimizing $\mathcal{L}_{cond}^{CDC}$ is equivalent to optimizing $\mathcal{L}_{marg}^{CDC}$.

**Theorem 3.1.** $\mathcal{L}_{cond}^{CDC}(\theta) = \mathcal{L}_{marg}^{CDC}(\theta) + C$, *where $C$ does not depend on $\theta$. Hence* $\nabla_\theta\mathcal{L}_{cond}^{CDC}(\theta) = \nabla_\theta\mathcal{L}_{marg}^{CDC}(\theta)$.

## 3.2. Mean-centered carré du champ matching

A similar argument applies in the the mean-centered case Eq. 6. Here, we additionally need to learn the terms $P_\varepsilon f$ and $P_\varepsilon h$ using other neural networks, which can be trained with the denoising loss $\mathbb{E}_{X,Y|X}\left[\left((P_\varepsilon f)^\theta(Y) - f(X)\right)^2\right]$. We then freeze the parameters and regress the mean-centered CDC using $\frac{1}{2\varepsilon}\left(f(\mathbf{x}) - (P_\varepsilon f)^\theta(\mathbf{y})\right)\left(f(\mathbf{x}) - (P_\varepsilon f)^\theta(\mathbf{y})\right)$ as the conditional CDC in Eq. 10.

## 3.3. Conditional Riemannian metric matching

The loss in Eq. 10 can be applied to any pair of scalar fields $f$ and $h$. However, in practice we will train a network to regress the CDC $\Gamma_\varepsilon(x_i, x_j)$ of each pair of coordinate functions $x_k : \mathcal{M} \to \mathbb{R}$ for $k = 1, \ldots, D$, and we show in Sec. 4.1 that the CDC $\Gamma_\varepsilon(f,h)$ of any pair $f, h : \mathcal{M} \to \mathbb{R}$ can be recovered from $\Gamma_\varepsilon(x_i, x_j)$ by the chain rule. Applying the conditional CDC loss jointly to all the coordinate functions $f = x_i$ and $h = x_j$ yields matrix-valued targets and predictions, and results in the aggregated Frobenius loss

$$\mathcal{L}_{cond}^{\text{Riem}} := \mathop{\mathbb{E}}_{X,Y|X}\left[\left\|\Gamma_\varepsilon^\theta(Y) - \frac{1}{2\varepsilon}(X-Y)(X-Y)^T\right\|_F^2\right], (11)$$

where $\Gamma_\varepsilon^\theta$ outputs a positive semi-definite $D \times D$ matrix approximating the CDC matrix. When using the mean-centered loss, we note that the term $(P_\varepsilon Y)^\theta$ becomes a standard denoising network (Li & He, 2025). We condition the network on $\varepsilon$, allowing it to capture the geometry of the data across different scales simultaneously. The full training algorithm can be found in Algorithm. 2. This loss is most related to Meng et al. (2021), who propose a loss to estimate the second order score $\nabla^2 \log p$ of the data. We discuss the exact relationship in Appendix B.

## 3.4. Positive semi-definite and low rank training

Since $\Gamma_\varepsilon(Y)$ is symmetric and positive semidefinite (PSD) by definition, we first parameterize a neural network to produce a $D \times D$ matrix $M_\varepsilon^\theta(Y)$ and make it symmetric PSD by setting $\Gamma_\varepsilon^\theta(Y) = M_\varepsilon^\theta(Y)^T M_\varepsilon^\theta(Y)$. Since $D$ can be much larger than the intrinsic dimension, we also consider a low rank version where $M_\varepsilon^\theta(Y) \in \mathbb{R}^{r \times D}$ making $\Gamma_\varepsilon(Y)$ symmetric PSD with a rank upper bounded by the hyperparameter $r$. In this case we observe that the Frobenius norm in Eq. 11 can be simplified by expanding

$$\left\|M_\varepsilon^\theta(Y)^T M_\varepsilon^\theta(Y) - \frac{1}{2\varepsilon}(X-Y)(X-Y)^T\right\|_F^2$$
$$= \left\|M_\varepsilon^\theta(Y)^T M_\varepsilon^\theta(Y)\right\|_F^2 + \frac{1}{4\varepsilon^2}\left\|X-Y\right\|^4$$
$$- \frac{1}{\varepsilon}\left\|M_\varepsilon^\theta(Y)(X-Y)\right\|^2.$$

Since $\left\|M_\varepsilon^\theta(Y)^T M_\varepsilon^\theta(Y)\right\|_F^2 = \left\|M_\varepsilon^\theta(Y)M_\varepsilon^\theta(Y)^T\right\|_F^2$, we can compute this loss without ever materializing any $D \times D$ matrices, considerably improving the memory and time complexity. The second term also does not depend on $\theta$, so we form the simplified low-rank loss

$$\mathcal{L}_{LR} = \mathop{\mathbb{E}}_{X,Y|X}\left[\left\|M_\varepsilon^\theta(Y)M_\varepsilon^\theta(Y)^T\right\|_F^2\right]$$
$$- \frac{1}{\varepsilon}\mathop{\mathbb{E}}_{X,Y|X}\left[\left\|M_\varepsilon^\theta(Y)(X-Y)\right\|^2\right] \quad (12)$$

If the manifold $\mathcal{M}$ has dimension $d$, we need to take $r$ to be at least $d$, but cannot generally assume that $r = d$ is enough

to fully factorize the metric (this would require, at least, the condition that $\mathcal{M}$ is *parallelizable*[1]). However, we prove in Sec. 4.2 that picking $r = 2d - 1$ is always enough for the low-rank factorization to be valid.

If a strict upper bound of $r$ to the intrinsic dimensionality is not desired, we consider the efficient parameterization $\Gamma_\varepsilon^\theta(Y) = M_\varepsilon^\theta(Y)^T M_\varepsilon^\theta(Y) + \lambda\mathbf{I}$ where $\lambda$ is a small Tikhonov regularization term ensuring strict positive definiteness. In this case, we get $\mathcal{L}_{LR}^\lambda = \mathcal{L}_{LR} + 2\lambda\|M_\varepsilon^\theta(Y)\|_F^2$, see Appendix C for a derivation.

## 4. Convergence and the Riemannian toolkit

For all $\varepsilon$, the CDC $\Gamma_\varepsilon$ defines a data-driven Riemannian metric which can be used to compute geometric objects via diffusion geometry. In this section, we provide a theoretical guarantee for our construction. Namely, when the data is locally sampled from a manifold and the loss is minimized, then $\Gamma_\varepsilon$ converges to the true metric as $\varepsilon \to 0$.

### 4.1. Recovering the metric from an embedded manifold

Our main theoretical guarantee is provided by the following theorem. We note that the manifold assumption only needs to hold locally around a point $x \in \mathcal{M}$, so the result extends to non-manifold data such as disjoint unions of manifolds or settings where the intrinsic dimension may vary across regions of the space. All proofs can be found in Appendix A.

**Theorem 4.1** (Convergence of the carré du champ)**.** *Let $\mu$ be a probability measure on $\mathbb{R}^D$ and $x \in supp(\mu)$. Suppose that $B(x, \delta) \cap supp(\mu)$ is a manifold (of any dimension, possibly depending on $x$), for some $\delta > 0$, with induced Riemannian metric $g$, and that $\mu$ has a smooth density on this manifold. Then $\Gamma_\varepsilon(f, h)(x) \to g_x(\nabla f, \nabla h)$ as $\varepsilon \to 0$, for all smooth functions $f, h$.*

**Carré du champ of the ambient coordinates converges to tangent space projection.** As a special case, we consider the ambient coordinate functions $x_k : \mathcal{M} \to \mathbb{R}$, $x_k(\mathbf{x}) = (\mathbf{x})_k$, for $k = 1, \ldots, D$. Applying the carré du champ to these functions yields, at each point $\mathbf{p} \in \mathcal{M}$, the matrix

$$(\mathbf{\Gamma}_\varepsilon(\mathbf{p}))_{k\ell} \approx g_\mathbf{p}(\nabla x_k, \nabla x_\ell),$$

where the approximation becomes exact when $\epsilon \to 0$. Let

$$G(\mathbf{p}) = (\Gamma(x_k, x_l)(\mathbf{p}))_{k\ell}$$

be the $D \times D$ carré du champ matrix of the ambient coordinates at $\mathbf{p}$, which corresponds to the pullback of the ambient Euclidean metric to the tangent space. Because

---

$\mathcal{M}$ is embedded in $\mathbb{R}^D$ *isometrically*, the eigenvalues of $G(\mathbf{p})$ are either 1 or 0, with eigenvectors pointing in the tangent and normal directions, respectively, and so $G(\mathbf{p})$ the projection matrix $\mathbb{R}^D \to T_\mathbf{p}\mathcal{M}$. We immediately obtain the following corollary by applying Theorem 4.1 to the entries of the matrix $\mathbf{\Gamma}_\varepsilon(\mathbf{p})_{k\ell}$.

**Corollary 4.2.** *Under the same assumptions as Theorem 4.1, the matrix $(\mathbf{\Gamma}_\varepsilon(\mathbf{p}))_{k\ell}$ converges (in every matrix norm) to the projection matrix onto the tangent space $\mathbb{R}^D \to T_\mathbf{p}\mathcal{M}$.*

**Carré du champ of the ambient coordinates is all you need.** The matrix $\mathbf{\Gamma}_\varepsilon(\mathbf{p})$ of the CDC of the ambient coordinates is important because it suffices to evaluate the CDC of arbitrary functions. If $f, h : \mathbb{R}^D \to \mathbb{R}$ are smooth, then

$$g_\mathbf{p}(\nabla f, \nabla h) = \sum_{k,l=1}^D \frac{\partial f}{\partial x_j}(\mathbf{p})\frac{\partial h}{\partial x_l}(\mathbf{p})g_\mathbf{p}(\nabla x_k, \nabla x_\ell)$$

by the chain rule, so

$$\Gamma(f, h)(\mathbf{p}) = \sum_{k,l=1}^D \frac{\partial f}{\partial x_k}(\mathbf{p})\frac{\partial h}{\partial x_l}(\mathbf{p})\Gamma(x_k, x_\ell)(\mathbf{p}).$$

If we denote the Jacobians of $f, h$ by $\boldsymbol{\partial} f(\mathbf{p})$ and $\boldsymbol{\partial} h(\mathbf{p})$, as in Eq. 1, then this is just the matrix conjugation

$$\Gamma(f, h)(\mathbf{p}) = \boldsymbol{\partial} f(\mathbf{p})^T \mathbf{\Gamma}_\varepsilon(\mathbf{p})\boldsymbol{\partial} h(\mathbf{p}). \tag{13}$$

We can formalize this in the following corollary.

**Corollary 4.3.** *Under the same assumptions as Theorem 4.1, we have $\boldsymbol{\partial} f(\mathbf{p})^T \mathbf{\Gamma}_\varepsilon(\mathbf{p})\boldsymbol{\partial} h(\mathbf{p}) \to g_\mathbf{p}(\nabla f, \nabla h)$ as $\varepsilon \to 0$, for all smooth $f, h$.*

### 4.2. Low-rank training is valid for $r \geq 2d - 1$

As discussed in Sec. 3.4, the low-rank factorization of the CDC can have topological obstructions if $r$ is too small. We now prove that, for at least $r \geq 2d - 1$, the minimizer of the low-rank loss will still converge to the correct metric $G(\mathbf{p})$.

**Theorem 4.4** (Low-rank training is valid for $r \geq 2d - 1$)**.** *If $r \geq 2d - 1$ and $M_\varepsilon^\theta$ minimises the low-rank loss (Eq. 12) then $M_\varepsilon^\theta(\mathbf{p})^T M_\varepsilon^\theta(\mathbf{p}) \to G(\mathbf{p})$ as $\varepsilon \to 0$.*

### 4.3. Intrinsic gradients

The carré du champ gives us direct access to the *intrinsic* gradients $\nabla f$ of functions $f$ defined on the data, with respect to the underlying geometry. If $\mathcal{M} \subseteq \mathbb{R}^D$ is an (isometrically embedded) submanifold , we can represent $\nabla f$ in ambient coordinates by its *pushforward* from $\mathcal{M}$ to $\mathbb{R}^D$. At a each point $\mathbf{p} \in \mathcal{M}$, this is given by the vector

$$(g_\mathbf{p}(\nabla f, \nabla x_1), ..., g_\mathbf{p}(\nabla f, \nabla x_D)) \in \mathbb{R}^D \tag{14}$$

where $g_{\mathbf{p}}(\nabla f, \nabla x_i)$ is the directional derivative of the function $x_i$ along $\nabla f$ at the point $\mathbf{p} \in \mathcal{M}$. By Eq. 2, these terms are just the CDC $g_{\mathbf{p}}(\nabla f, \nabla x_i) = \Gamma(f, x_i)$. When $f : \mathbb{R}^D \rightarrow \mathbb{R}$ is defined on the ambient space (such as a neural network or an ambient potential), we can apply Eq. 13 and the fact that $\boldsymbol{\partial} x_i(\mathbf{p}) = \mathbf{e}_i$, to write

$$
\begin{aligned}
\Gamma(f, x_i)(\mathbf{p}) &= \boldsymbol{\partial} x_i(\mathbf{p})^T \boldsymbol{\Gamma}_\varepsilon(\mathbf{p}) \boldsymbol{\partial} f(\mathbf{p}) \\
&= (\boldsymbol{\Gamma}_\varepsilon(\mathbf{p}) \boldsymbol{\partial} f(\mathbf{p}))_i
\end{aligned}
\tag{15}
$$

for each $i = 1, ..., D$. Substituting Eq. 15 into Eq. 14, we can compute the intrinsic gradient $\nabla f$ by the simple matrix-vector product $\boldsymbol{\Gamma}_\varepsilon(\mathbf{p}) \boldsymbol{\partial} f(\mathbf{p}) \in \mathbb{R}^D$. We apply this method in the middle column of Fig. 1.

### 4.4. Riemannian optimization and interpolating paths

One application of the Riemannian metric is to perform *Riemannian optimization*, in which we aim to optimize a function $\mathcal{M} \rightarrow \mathbb{R}$ by gradient descent while remaining on the manifold $\mathcal{M}$. We can formulate this problem using the carré du champ (thereby also generalising it to non-manifold geometries), as follows.

In general, when $X$ is a vector field on an (isometrically embedded) submanifold $\mathcal{M} \subseteq \mathbb{R}^D$, and $\gamma : [0,1] \rightarrow \mathcal{M}$ is a flow line (integral curve) of $X$, then $\gamma$ has gradient $\dot{\gamma}(t) = X(\gamma(t))$. When $X = \nabla f$ is the gradient of some function $f : \mathcal{M} \rightarrow \mathbb{R}$, then $X(\gamma(t))$ is given by Eq. 14, so we can compute the flow along $\nabla f$ by solving the ODE

$$
\dot{\gamma}(t) = \boldsymbol{\Gamma}_\varepsilon(\mathbf{p}) \boldsymbol{\partial} f(\mathbf{p}).
\tag{16}
$$

We notice that, in the case that $\mathcal{M} = \mathbb{R}^D$ and Eq. 16 is discretized via an Euler method, then get

$$
\gamma(t+1) = \gamma(t) - \eta_t G(\gamma(t))^{-1} \boldsymbol{\partial} f(\gamma(t)),
$$

where $\eta_t$ is a learning rate that may depend on $t$, and $G(\mathbf{p}) = \boldsymbol{\Gamma}_\varepsilon(\mathbf{p})^{-1}$ is the inverse of the carré du champ.

We apply this method to find interpolating paths between a source point $\mathbf{x}_0$ and a target point $\mathbf{p}$ that stay close to the data manifold. We aim to minimize the function $f(\mathbf{x}) = \frac{1}{2}\|\mathbf{x} - \mathbf{p}\|^2$ with initial condition $\mathbf{x}_0$, so the interpolating path is the trajectory of $\mathbf{x}_t$. When the Riemannian metric is the standard Euclidean metric, the resulting trajectory will just be the linear interpolation path in $\mathbb{R}^D$. When the Riemannian metric is different, then these paths are not always guaranteed to converge to $\mathbf{p}$ or be the shortest paths possible.

## 5. Experiments

We apply metric matching on synthetic and real-world datasets to evaluate i) its scalability to large datasets, ii) its accuracy in controlled settings with ground truth, and iii)

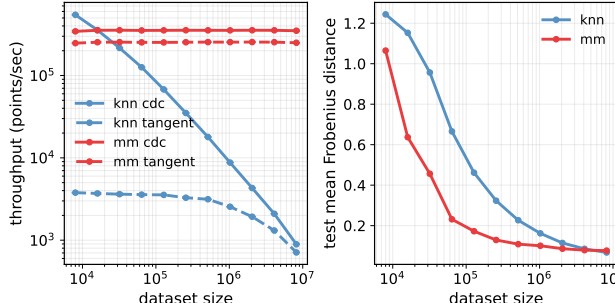

*Figure 2.* Comparison of trained metric matching (mm) network against $k$-NN-based baseline, on throughput (left) and performance (right). All experiments were run on a NVIDIA A10 (24GB) GPU.

its scalability to high dimensions. First, we use synthetic datasets with known ground-truth geometry to assess the accuracy of intrinsic dimension and tangent space estimation, as well as the scalability of the method with increasing sample size. Second, we apply our approach to a high dimensional image dataset to demonstrate that it can be trained and evaluated at scales where classical geometric methods based on $k$-nearest neighbor graphs become impractical.

In each dataset, we estimate the intrinsic dimensionality and local tangent spaces. Following Jones (2024b), both quantities are derived from the predicted CDC matrix (usually computed via a $k$-NN graph). Intrinsic dimension is estimated from the eigenvalues of $\Gamma_\varepsilon(\mathbf{p})$, while the tangent space (when $d$ is known) is given by the eigenvectors associated with its top $d$ eigenvalues. We provide additional experimental details in Appendix E.

### 5.1. Synthetic setting: scalability and accuracy

We begin with a controlled synthetic experiment on $N$ points sampled uniformly from the $d$-dimensional sphere embedded in $\mathbb{R}^D$. We vary the number of samples $N$ to evaluate the scalability of metric matching relative to baselines. We also use the ground-truth tangent spaces to quantitatively evaluate tangent space prediction. Unless otherwise specified, we use $d = 8$, $D = 64$, and a fixed architecture across all dataset sizes: an MLP with 15M parameters predicting a rank $r = 16$ matrix via low-rank training.

**Scalability.** We first use this setup to evaluate the scalability of the neural surrogate for CDC-based geometric estimators against $k$-NN-based CDC estimators, and report throughput (number of inferences per second) as a function of the number of samples. We compare both the raw CDC computation, and eigen-decomposition to identify the leading $d$ eigenvectors, as is typical in tangent space prediction. The raw neural CDC surrogate becomes advantageous over the raw CDC estimator at approximately 16k points, and is over $82\times$ faster for 2M points and $400\times$ faster for 8M points. In the small-data regime, the dominant computational bottleneck for tangent space prediction is eigendecom-

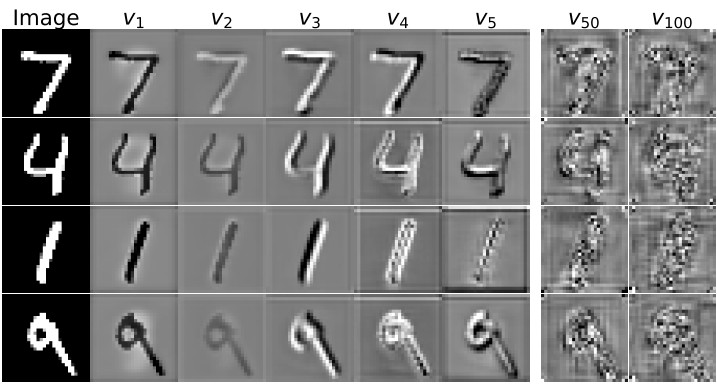
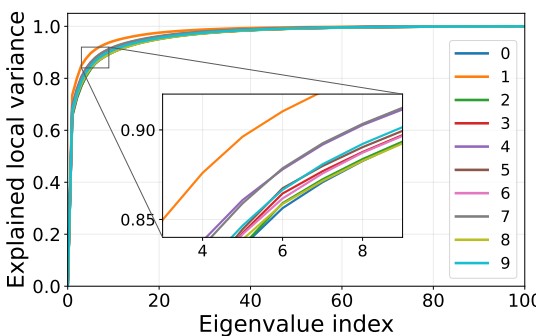

*(a)* Visualization of the first 5, 50th, and 100th eigenvectors of the CDC matrix learned by metric matching.

*(b)* Class-wise mean cumulative eigenvalues.

*Figure 3.* MNIST CDC eigen-analysis.

position: low-rank training substantially reduces this cost by replacing a $D \times D$ eigendecomposition with an $r \times r$ one, where $r \ll D$, leading to speedups of $67\times$, $132\times$, $194\times$, and $359\times$ for 8k, 2M, 4M, and 8M points, respectively.

**Accuracy.** Next, we evaluate the accuracy of tangent space prediction by measuring the Frobenius distance $\|UU^T - \hat{U}\hat{U}^T\|_F$ on a test set, where $U \in \mathbb{R}^{d \times D}$ denotes a basis for the ground-truth tangent space and $\hat{U}$ is the estimator obtained by eigendecomposition. The neural surrogate clearly outperforms $k$-NN for datasets smaller than 4M, with $46\%$ improved performance relative to the $k$-NN baseline, while closely matching it for larger datasets. We note that the performance gains saturate for very large datasets, which we attribute to current architectural choices, which could be improved with further tuning. We visualize the eigenvalues in Fig. 7, where the neural surrogate accurately captures 8 leading dimensions, while the $k$-NN CDC captures 9. These results indicate that the neural surrogate demonstrates better generalization than the $k$-NN–based CDC.

### 5.2. High-dimensional image datasets

Classical kernel methods tend to be cursed by dimension, restricting their scalability to high-dimensional datasets. We test metric matching in this setting by attempting to recover the intrinsic geometry of image datasets.

**Geometric analysis of MNIST.** We apply metric matching to study the 784 dimensional image dataset MNIST, consisting of 60k training and 10k validation images, which are black and white images representing digits 0 to 9. We train a standard UNet architecture with the low-rank metric matching objective Eq. 12 with rank 100. We visualize the quality of the learned geometry by plotting the eigenvectors of the CDC at validation images in Fig. 3a. We see that the first eigenvectors are highly interpretable, compared to the 50th and 100th which appear much noisier. In particular, the second eigenvector is close to a damped version of the

original image, while the third and fourth are concentrated at edges and so generate translations. We evaluate the importance of each tangent direction by its eigenvalue, and we plot the cumulative eigenvalue distribution (i.e. local explained variance) for each class in Fig. 3b. We observe a sharp dropoff in eigenvalues, reflecting the low-dimensional structure. The digit 1 has a faster drop, suggesting a lower intrinsic dimension, which is perhaps due to its relative visual simplicity.

**Evaluating metric quality in high dimensions.** We next apply metric matching to learn the carré du champ on three additional datasets: CIFAR-10 (60k colored $32 \times 32$ images, $D = 3072$), celebA (200k colored $64 \times 64$, $D = 12288$), and FFHQ (70k colored $256 \times 256$, $D = 196608$). These examples lack a ground truth for evaluating and comparing models, so we introduce the following metric based on inception feature stability, inspired by the *Frechet Inception Distance* (FID) (Heusel et al., 2017) for evaluating generative models. We compute features from a standard pretrained inception network $\Phi$ used for FID, and consider how those features vary when the input image is perturbed. If we interpret $\Gamma_x$ as a Gaussian covariance matrix for the tangent space at $x$, and pick a target perturbation magnitude $\alpha$, we can sample $v \sim \mathcal{N}(x, \Gamma_x)$, renormalize $\|v\| = \alpha$, and compute the perturbation $\|\Phi(x + v) - \Phi(x)\|$. When the perturbation is tangent to the manifold, the difference in features should be small (Srinivas et al., 2023), so we average the perturbation over $v$ and $x$ to get a quality measure for $\Gamma$ at each magnitude $\alpha$, where no ground truth exists. Normalization of the perturbation magnitude is important to avoid a bias towards low-rank metrics: if all the tangent vectors have the same magnitude, we measure the quality of *direction only*. This evaluation is like the local version of the *perceptual path length* (Karras et al., 2019).

We plot the mean and standard deviation bars over the dataset, for each $\alpha$, as curves in Fig. 4. As a baseline, we add perturbation in random directions, which corresponds

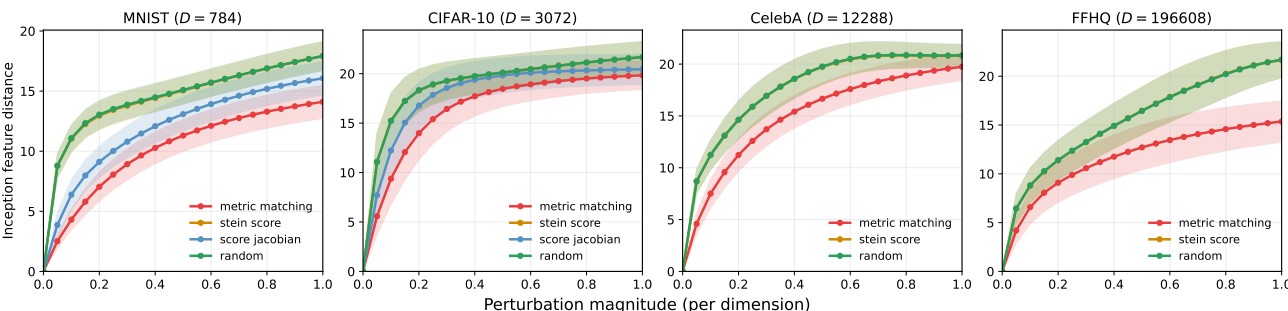

*Figure 4.* Inception feature stability under tangent perturbation.

| Method | Metric | MNIST | CIFAR-10 | CelebA | FFHQ |
|---|---|---|---|---|---|
| Stein Score | Time | 0.9±0.0 | 17.0±0.4 | 6.5±1.8 | 21.5±0.3 |
| | Memory | 2.5 MB | 8.8 MB | 31.6 MB | 444.3 MB |
| Score Jacobian | Time | 75.5±7.1 | 992.3±1.2 | – | – |
| | Memory | 2.7 GB | 27.4 GB | OOM | OOM |
| Metric Matching | Time | 1.5±2.6 | 16.6±3.7 | 6.2±1.9 | 21.9±4.4 |
| | Memory | 2.7 MB | 8.8 MB | 17.0 MB | 513.9 MB |

*Table 1.* Runtime and memory comparison on a single NVIDIA H100 GPU. Times are reported in milliseconds.

to the Euclidean (isotropic) metric. We also ablate metric matching against the score Jacobian metric (Kharitenko et al., 2025) and Stein score metric (Azeglio & Di Bernardo, 2025). The Stein score metric is a rough approximation given by a rank-1 correction to the ambient identity matrix, whose effect is pronounced in lower dimensions but vanishes as dimension increases (we see a slight improvement on randomness for MNIST and CIFAR but virtually no gain on CelebA and FFHQ). The score Jacobian is also a principled estimator for the true geometry, and performs much better, but underperforms metric matching. We expect that this is because its metric is full-rank, without the regularization effect of low-rank training of metric matching. The results do not depend sensitively on bandwidth, and we report the best result for each method over a grid search.

We also report the time and memory requirements of each method in Tab. 1. An additional practical constraint is the high computational cost of computing the Jacobian, which requires $\sim 3000\times$ more memory and is $\sim 62\times$ slower per evaluation on CIFAR-10, and results in OOM on CelebA and FFHQ (despite 80GB GPU memory). Conversely, the Stein score metric and metric matching have low-rank parametrizations so scale easily to high dimensions.

### 5.3. Interpolation along the data manifold

The tangent vectors visualized in Fig. 3a demonstrate that metric matching can learn a meaningful data geometry at individual points. To test whether this geometry is globally consistent, we apply the learned CDC matrix to find interpolating paths along the data manifold via Riemannian optimization, following the method described in Sec. 4.4.

We apply this method to synthetic data in the bottom right of Fig. 1. We then apply it to find paths between pairs of MNIST digits in the same class, which we visualize in Fig. 5, and compare them to linearly interpolated paths (LERP) in the ambient space $\mathbb{R}^D$. In each case, the intrinsic paths obtained from the metric matching CDC show smooth deformations from one digit to the other, with none of the 'ghost' features of the LERP paths, showing that geometry learned by metric matching is consistent across the whole data geometry. Metric matching can also be applied in latent space, which allows comparison to methods learning a Riemannian metric in the latent space of a VAE (Arvanitidis et al., 2020), we show the results in Fig. 9.

## 6. Related work

**Geometry in diffusion and generative models.** Recent work has highlighted deep connections between diffusion models and differential geometry. Stanczuk et al. (2024) show that, at small noise levels $\sigma$, the score $\nabla \log p_\sigma$ aligns with directions normal to the data manifold, and exploit this for intrinsic dimension estimation. Kadkhodaie et al. (2024) study the generalization properties of denoisers through the eigendecomposition of the Jacobian of the score model (the Hessian $\nabla^2 \log p_\sigma$ of the log-density), revealing a rapid spectral decay and semantically meaningful leading eigenvectors. More recently, Kharitenko et al. (2025) formalize this connection by showing that $\mathbf{I} + \sigma^2 \nabla^2 \log p_\sigma$ converges to the tangent space projector, which can be accessed via automatic differentiation of the score, and apply this to Riemannian optimization on data manifolds. Our work complements and extends this line of research by explicitly linking modern denoising objectives to classical diffusion maps (Coifman & Lafon, 2006). Our denoising-style loss directly regresses the carré du champ, providing the data-dependent Riemannian metric through a single forward pass while also guaranteeing convergence to the tangent-space projector.

**(Riemannian) metric learning.** Metric learning originally aimed to learn task-adapted distances for $k$-NN classification and clustering, extending beyond the standard Euclidean metric (Friedman, 1994; Xing et al., 2002; Bel-

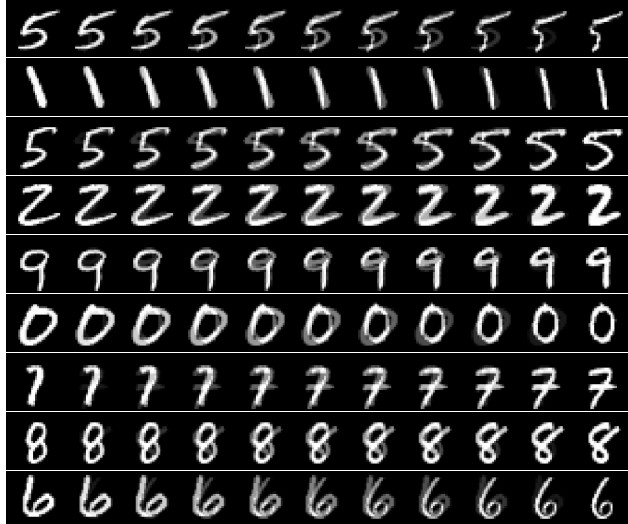

*(a)* Linear interpolation (LERP).

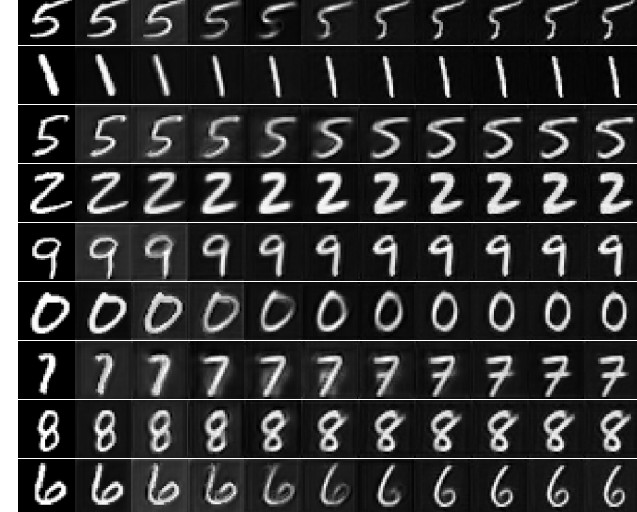

*(b)* Manifold interpolation.

*Figure 5.* Comparison of linear and manifold interpolation trajectories, sampled uniformly along the interpolation path.

let et al., 2015). More recently, there has been a surge of interest in a Riemannian formulation, which would additionally permit intrinsic geometric notions such as geodesics and curvature. Most such methods either attempt to learn the metric of the embedded data manifold or to design a metric that is meaningful for a downstream task. Metric learning methods can be broadly grouped into unsupervised, supervised, and implicit approaches. Unsupervised methods infer geometry from samples via graphs or kernels methods, e.g. (Jones & Lanners, 2026; Arvanitidis et al., 2016). Supervised methods use a parametric model trained using a loss function based on labels, such as contrastive (Huang et al., 2014) or triplet losses (Weinberger & Saul, 2009). Implicit methods define geometry as a by-product of a learned model, e.g. via pullback metrics induced by the Jacobian of a VAE or embedding model (Arvanitidis et al., 2018; Diepeveen et al., 2025). This Riemannian approach has recently been applied to generative modeling (Kapusniak et al., 2024; Bamberger et al., 2026) and data analysis (Diepeveen et al., 2024), extending beyond its original use in clustering (Xing et al., 2002) and classification (Friedman, 1994). We refer to Gruffaz & Sassen (2025) for a comprehensive survey. Our work builds on this literature by introducing a self-supervised objective for learning the intrinsic geometry of data manifolds, with theoretical guarantees at optimality.

**Riemannian optimization on the data manifold.** Riemannian optimization was originally introduced for minimizing objectives subject to constraints that form a smooth manifold, allowing the optimization problem to be expressed intrinsically using Riemannian geometry (Absil et al., 2008; Boumal, 2023). Most existing algorithms assume that the manifold is given explicitly, along with its associated geometric operations, like the exponential or vector transport maps. In contrast, the geometry of data manifolds is only im-

plicit in finite samples, and there is typically no parametrization or scalable statistical estimate for the required operations. While this problem has received renewed attention in recent work (Diepeveen & Weber, 2025; Kharitenko et al., 2025), scalable and principled optimization methods in this sample-based regime remain relatively underdeveloped.

## 7. Conclusion

We presented metric matching: a tractable denoising-style objective for estimating the carré du champ with neural networks. This provides a scalable alternative to graph CDC estimators, enabling out-of-sample geometry prediction in a single forward pass. On synthetic manifolds, metric matching improves the accuracy and throughput compared to classical $k$-NN baselines. On high-dimensional image data, metric matching remains effective where $k$-NN methods fail due to the curse of dimensionality, thereby expanding the practical reach of diffusion geometry tools to high-dimensional data. We hypothesize that this robustness stems from the inductive biases of the neural estimator: characterizing when and why such generalization occurs is an important direction for future work. We hope that this work can provide a platform for developing novel deep learning methods that directly inherit tools from Riemannian geometry.

## Acknowledgements

We thank Romeo Passaro for useful discussions on connections between metric matching and higher order score matching. This work is partially supported by the EPSRC Turing AI World-Leading Research Fellowship No. EP/X040062/1 and EPSRC AI Hub No. EP/Y028872/1. AG was supported by an ERC grant (NEURO-FUSE, Project DOI: 10.3030/101163046).

## Impact statement

This paper presents work whose goal is to advance the field of machine learning. There are many potential societal consequences of our work, none of which we feel must be specifically highlighted here.

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

## A. Proofs

**Theorem 3.1.** $\mathcal{L}_{cond}^{CDC}(\theta) = \mathcal{L}_{marg}^{CDC}(\theta) + C$, *where $C$ does not depend on $\theta$. Hence $\nabla_\theta \mathcal{L}_{cond}^{CDC}(\theta) = \nabla_\theta \mathcal{L}_{marg}^{CDC}(\theta)$.*

*Proof.* Let $Z := \Gamma_\varepsilon(f, h)(X, Y)$ be the random variable defined by $X \sim p$ and $Y \mid X \sim \mathcal{N}(X, \varepsilon I)$, so Eq. 9 says that $\mathbb{E}[Z \mid Y] = \Gamma_\varepsilon(f, h)(Y)$. We can apply the law of total expectation and bias-variance decomposition to derive

$$
\begin{aligned}
\mathcal{L}_{cond}^{CDC}(\theta) &= \mathbb{E}\left[(\Gamma_\varepsilon^\theta(Y) - Z)^2\right] \\
&= \mathbb{E}\left[\mathbb{E}\left[(\Gamma_\varepsilon^\theta(Y) - Z)^2 \mid Y\right]\right] \\
&= \mathbb{E}\left[\left(\Gamma_\varepsilon^\theta(Y) - \mathbb{E}[Z \mid Y]\right)^2 + \mathrm{Var}(Z \mid Y)\right] \\
&= \mathbb{E}\left[\left(\Gamma_\varepsilon^\theta(Y) - \Gamma_\varepsilon(f, h)(Y)\right)^2\right] + \mathbb{E}\left[\mathrm{Var}(Z \mid Y)\right] \\
&= \mathcal{L}_{marg}^{CDC}(\theta) + \mathbb{E}\left[\mathrm{Var}(Z \mid Y)\right],
\end{aligned}
$$

so the result holds with $C = \mathbb{E}\left[\mathrm{Var}(Z \mid Y)\right]$. $\qquad \square$

**Theorem 4.1** (Convergence of the carré du champ). *Let $\mu$ be a probability measure on $\mathbb{R}^D$ and $x \in supp(\mu)$. Suppose that $B(x, \delta) \cap supp(\mu)$ is a manifold (of any dimension, possibly depending on $x$), for some $\delta > 0$, with induced Riemannian metric $g$, and that $\mu$ has a smooth density on this manifold. Then $\Gamma_\varepsilon(f, h)(x) \to g_x(\nabla f, \nabla h)$ as $\varepsilon \to 0$, for all smooth functions $f, h$.*

*Proof.* We would first like to show that, by possibly reducing $\delta$ to any $0 < \delta' < \delta$, we can assume that $\mathcal{M} = B(x, \delta) \cap supp(\mu)$ is a compact manifold with boundary. By assumption, $\mathcal{M}$ is a smooth manifold of $\mathbb{R}^D$, so the ambient distance function $y \mapsto \|y - x\|$ restricts to a smooth function $d_x : \mathcal{M} \to \mathbb{R}$. In particular, $\mathcal{M} = d_x^{-1}[0, \delta)$, and $\mathcal{M}' = d_x^{-1}[0, \delta']$ is a compact submanifold of $\mathbb{R}^D$ with boundary (that is also a submanifold of $\mathcal{M}$), and satisfies all the other conditions on $\mathcal{M}$. We can now apply the *diffusion maps* theorem (Coifman & Lafon, 2006; Belkin & Niyogi, 2003) to the operator $\mathcal{L}_\epsilon := (\mathbf{I} - P_\varepsilon)/\varepsilon$ on $\mathcal{M}$. Let us write $g$ for the induced Riemannian metric on $\mathcal{M}$, $\Delta_g$ for its Laplace-Beltrami operator, and $p$ for the smooth density of $\mu$ with respect to the Riemannian volume form of $(\mathcal{M}, g)$. Then, for smooth $f$, Theorem 2 in (Coifman & Lafon, 2006) (applied here with $\alpha = 0$) states that $|\mathcal{L}_\epsilon f - \mathcal{L}f| \to 0$ as $\varepsilon \to 0$, where

$$
\mathcal{L}f = \frac{\Delta_g(fp) - \Delta_g(p)f}{p}.
$$

We can apply the carré du champ identity Eq. 2 to write

$$
\Delta_g(fp) - \Delta_g(p)f = \Delta_g(f)p - 2g(\nabla p, \nabla f),
$$

so

$$
\mathcal{L}f = \Delta_g f - 2\frac{g(\nabla p, \nabla f)}{p} = \Delta_g f - 2g(\nabla \log p, \nabla f).
$$

We can now again use the carré du champ identity to expand

$$
|\Gamma_\varepsilon(f, g) - g_x(\nabla f, \nabla h)| = \frac{1}{2}\left|f(\mathcal{L}_\varepsilon h - \Delta_g h) + h(\mathcal{L}_\varepsilon f - \Delta_g f) - (\mathcal{L}_\varepsilon(fh) - \Delta_g(fh))\right| \tag{17}
$$

and notice that each term satisfies

$$
\mathcal{L}_\varepsilon h - \Delta_g h = \mathcal{L}_\varepsilon h - \mathcal{L}h + \mathcal{L}h - \Delta_g h = (\mathcal{L}_\varepsilon h - \mathcal{L}h) - 2g(\nabla \log p, \nabla h),
$$

so Eq. 17 becomes

$$
\frac{1}{2}\left|f[\mathcal{L}_\varepsilon h - \mathcal{L}h] + h[\mathcal{L}_\varepsilon f - \mathcal{L}f] - [\mathcal{L}_\varepsilon(fh) - \mathcal{L}(fh)] - 2fg(\nabla \log p, \nabla h) - 2hg(\nabla \log p, \nabla f) + 2g(\nabla \log p, \nabla(fh))\right|.
$$

We can apply the Leibniz rule to

$$
-2fg(\nabla \log p, \nabla h) - 2hg(\nabla \log p, \nabla f) + 2g(\nabla \log p, \nabla(fh)) = 2g(\nabla \log p, \nabla(fh) - f\nabla h - h\nabla f) = 0,
$$

and so find that

$$|\Gamma_\varepsilon(f,g) - g_x(\nabla f, \nabla h)| \leq \frac{1}{2}\big(|f||\mathcal{L}_\varepsilon h - \mathcal{L}h| + |h||\mathcal{L}_\varepsilon f - \mathcal{L}f| + |\mathcal{L}_\varepsilon(fh) - \mathcal{L}(fh)|\big) \to 0$$

as $\varepsilon \to 0$, since $|\mathcal{L}_\epsilon f - \mathcal{L}f| \to 0$ and $f, h$ are bounded (which is implied by the fact that $\mathcal{M}$ is compact). $\qquad\square$

**Lemma A.1** (Existence of a low-rank factorization if $K = 2d - 1$). *Let* $G(\mathbf{p}) = (\Gamma(x_i, x_j)(\mathbf{p}))_{i,j=1,\ldots,D}$ *be the* $D \times D$ *carré du champ matrix of the ambient coordinates at* $\mathbf{p}$. *Then, for* $K = 2d - 1$, *there exists a* $K \times D$ *matrix* $M(\mathbf{p})$, *whose entries are all smooth functions, such that* $G(\mathbf{p}) = M(\mathbf{p})^T M(\mathbf{p})$ *for all* $\mathbf{p} \in \mathcal{M}$.

*Proof.* Let $\mathcal{M}$ be a smooth $d$-dimensional manifold. If $d = 1$, the tangent bundle is trivial, so the result follows directly with $K = 1$. If $d \geq 2$, Whitney's immersion theorem gives a smooth immersion $\phi : \mathcal{M} \to \mathbb{R}^{2d-1}$. By definition, this means that the differential $d\phi : T\mathcal{M} \to T\mathbb{R}^K \cong \mathbb{R}^K \times \mathbb{R}^K$ defines an injective map $d\phi_p : T_p\mathcal{M} \to \mathbb{R}^K$ for all $\mathbf{p} \in \mathcal{M}$. In other words, $d\phi$ is a smooth embedding of vector bundles $T\mathcal{M} \to \varepsilon^K$, where $\varepsilon^K$ is the trivial bundle of rank $K$. We would like this derivative to be an isometry on each tangent space, i.e. that

$$g_p(u, v) = \langle d\phi_p(u), d\phi_p(v) \rangle_{\mathbb{R}^K}$$

for all $u, v \in T_p\mathcal{M}$, and we will now construct a smooth bundle map $\Phi : T\mathcal{M} \to \varepsilon^K$ that satisfies this.

Let us define a new Riemannian metric $h_\mathbf{p}$ by

$$h_p(u, v) = \langle d\phi_p(u), d\phi_p(v) \rangle_{\mathbb{R}^K}$$

for $u, v \in T_p\mathcal{M}$, which is strictly positive definite because $d\phi$ is injective. Since $\phi$ is smooth and $d\phi$ is injective for all $\mathbf{p}$, there exists a unique smooth section of the bundle of symmetric endomorphisms, $A \in \Gamma(\mathrm{End}(T\mathcal{M}))$, such that

$$g_p(A_p u, v) = h_p(u, v)$$

for all $u, v \in T_p\mathcal{M}$, which we can think of as the matrix representation of the quadratic form $h_\mathbf{p}$ with respect to the inner product $g_\mathbf{p}$. Both $g$ and $h$ are positive-definite inner products, so the operator $A_p$ must be symmetric and positive-definite with respect to $g$ for all $p \in M$.

We would now like to compute the inverse square root $A^{-1/2}$ as another smooth section in $\Gamma(\mathrm{End}(T\mathcal{M}))$. Since $A_p$ is positive-definite, its eigenvalues are strictly positive real numbers. The function $f(x) = x^{-1/2}$ is smooth on the interval $(0, \infty)$ and so, by the spectral theorem and standard functional calculus, the operator field $A^{-1/2}$ is a well-defined, smooth section of $\mathrm{End}(T\mathcal{M})$.

We now define a new bundle map $\Psi : T\mathcal{M} \to \mathbb{R}^K$ by pre-composing the immersion differential with the correction operator

$$\Psi_p := d\phi_p \circ A_p^{-1/2},$$

which is an isometry since

$$\begin{aligned}
\langle \Psi_p u, \Psi_p v \rangle_{\mathbb{R}^K} &= \langle d\phi_p(A_p^{-1/2}u), d\phi_p(A_p^{-1/2}v) \rangle_{\mathbb{R}^K} \\
&= h_p(A_p^{-1/2}u, A_p^{-1/2}v) \\
&= g_p(A_p(A_p^{-1/2}u), A_p^{-1/2}v) \\
&= g_p(A_p^{-1/2}A_p A_p^{-1/2}u, v) \\
&= g_p(u, v)
\end{aligned}$$

for all $u, v \in T_p\mathcal{M}$. We can now construct the matrix $K \times D$ factor $M(\mathbf{p})$ using the $K$ components of this bundle map. If $e_1, \ldots, e_K$ are the standard basis of $\mathbb{R}^K$, and $x_1, \ldots, x_D : \mathcal{M} \to \mathbb{R}$ are the ambient coordinate functions of $\mathcal{M}$ in $\mathbb{R}^D$, then we can set

$$M_{ij} = \langle e_i, \Psi(\nabla x_j) \rangle_{\mathbb{R}^K},$$

which are all smooth functions, and find that

$$
\begin{aligned}
(M^T M)_{ij} &= \sum_{r=1}^{K} M_{ri} M_{rj} \\
&= \sum_{r=1}^{K} \langle e_r, \Psi(\nabla x_i) \rangle \langle e_r, \Psi(\nabla x_j) \rangle \\
&= \langle \Psi(\nabla x_i), \Psi(\nabla x_j) \rangle_{\mathbb{R}^K} \\
&= g(\nabla x_i, \nabla x_j)
\end{aligned}
$$

for each $i, j = 1, ..., D$. $\qquad\square$

The Lemma demonstrates that the low-rank loss and the metric matching loss have the same minimzer for $K \geq 2d - 1$, and so we immediately attain the following theorem as a consequence.

**Theorem 4.4** (Low-rank training is valid for $r \geq 2d - 1$). *If $r \geq 2d - 1$ and $M_\varepsilon^\theta$ minimises the low-rank loss (Eq. 12) then $M_\varepsilon^\theta(\mathbf{p})^T M_\varepsilon^\theta(\mathbf{p}) \to G(\mathbf{p})$ as $\varepsilon \to 0$.*

## B. Extended comparison to higher order score matching.

We relate our conditional CDC objective to higher-order score matching (Meng et al., 2021), that propose a method to learn the higher order scores $s_k(x) = \nabla^k \log p_\varepsilon(x)$ for order $k$, where $p_\varepsilon = p_{\text{data}} \star \mathcal{N}(0, \varepsilon I)$. In the second order case they parameterize a first-order score network $s_1^\theta$ and a second-order network $s_2^\theta$, trained jointly via score matching and a second-order regression loss of the form

$$
\mathbb{E}_{X,Y|X} \left\| s_2^\theta(Y) + s_1^\theta(Y) s_1^\theta(Y)^\top + \frac{I - \frac{1}{\varepsilon}(X-Y)(X-Y)^\top}{\varepsilon} \right\|_F^2 .
$$

Let $A^*(y) := s_2(y) + s_1(y) s_1(y)^\top$. Optimizing the above yields the conditional identity

$$
A^*(y) = \mathbb{E}\left[ \frac{I - \frac{1}{\varepsilon}(X-Y)(X-Y)^\top}{\varepsilon} \,\middle|\, Y = y \right].
$$

With our definition $\Gamma^*(y) := \mathbb{E}\left[ \frac{(X-Y)(X-Y)^\top}{2\varepsilon} \,\middle|\, Y = y \right]$, we obtain the exact relationship $\varepsilon A^*(y) = I - 2\Gamma^*(y)$. While Meng et al. (2021) do not discuss convergence as $\varepsilon \to 0$ under manifold-supported data, we observe that $\Gamma^*(y)$ remains bounded and converges to the tangent-space projector, while $A^*(y)$ (and hence $s_2$) typically contains $\mathcal{O}(1/\varepsilon)$ components in directions normal to the manifold and can therefore diverge as $\varepsilon \to 0$. Additionally, (Meng et al., 2021) do not use the low-rank loss decomposition discussed in Sec. 3.4 which reduces the loss' complexity from $\mathcal{O}(D^2)$ to $\mathcal{O}(D)$, which is crucial to applying the loss to high dimensional images. Recently, Kharitenko et al. (2025) showed that $\mathbf{I} - \varepsilon s_2$ converges to the projection on the tangent space of the manifold as $\varepsilon \to 0$. Note that this is consistent with our results: $\varepsilon A^* = \varepsilon s_2 + \varepsilon s_1 s_1^T$, the $\varepsilon s_1$ term dies because $s_1$ remains bounded on the manifold, leaving only the $\varepsilon s_2$ term, and both converge to the same object.

## C. Low rank loss derivation

When the metric is parameterized as low rank with small identity perturbation $\Gamma_\varepsilon^\theta = M_\varepsilon^{\theta^T} M_\varepsilon^\theta + \lambda \mathbf{I}$, and let $\Delta = Y - X \sim \mathcal{N}(0, \varepsilon \mathbf{I})$ be the noise, then:

$$\left\| M_\varepsilon^{\theta^T} M_\varepsilon^\theta + \lambda \mathbf{I} - \frac{1}{2\varepsilon} \Delta \Delta^T \right\|_F^2 = \operatorname{Tr}\left( \left( M_\varepsilon^{\theta^T} M_\varepsilon^\theta + \lambda \mathbf{I} - \frac{1}{2\varepsilon} \Delta \Delta^T \right)^2 \right) \tag{18}$$

$$= \operatorname{Tr}\left( \left( M_\varepsilon^{\theta^T} M_\varepsilon^\theta + \lambda \mathbf{I} \right)^2 + \left( \frac{1}{2\varepsilon} \Delta \Delta^T \right)^2 - \frac{1}{\varepsilon} \left( M_\varepsilon^{\theta^T} M_\varepsilon^\theta + \lambda \mathbf{I} \right) \Delta \Delta^T \right) \tag{19}$$

$$= \operatorname{Tr}\left( \left( M_\varepsilon^{\theta^T} M_\varepsilon^\theta \right)^2 \right) + 2\lambda \operatorname{Tr}\left( M_\varepsilon^{\theta^T} M_\varepsilon^\theta \right) + \lambda^2 D + \operatorname{Tr}\left( \left( \frac{1}{2\varepsilon} \Delta \Delta^T \right)^2 \right) \tag{20}$$

$$- \frac{1}{\varepsilon} \left( \operatorname{Tr}\left( M_\varepsilon^{\theta^T} M_\varepsilon^\theta \Delta \Delta^T \right) + \lambda \operatorname{Tr}\left( \Delta \Delta^T \right) \right). \tag{21}$$

Where we expanded the squares. Now we notice that $\operatorname{Tr}\left( (\Delta \Delta^T)^2 \right) = \operatorname{Tr}\left( \Delta \Delta^T \Delta \Delta^T \right) = \|\Delta\|_2^2 \operatorname{Tr}\left( \Delta \Delta^T \right) = \|\Delta\|_2^4$, and due to the cyclic property of the trace (i.e. $\operatorname{Tr}(ABC) = \operatorname{Tr}(CAB)$), we have $\operatorname{Tr}\left( M_\varepsilon^{\theta^T} M_\varepsilon^\theta \Delta \Delta^T \right) = \operatorname{Tr}\left( \Delta^T M_\varepsilon^{\theta^T} M_\varepsilon^\theta \Delta \right) = \operatorname{Tr}\left( \left( M_\varepsilon^\theta \Delta \right)^T M_\varepsilon^\theta \Delta \right) = \|M_\varepsilon^\theta \Delta\|_2^2$. Reorganizing the terms we get:

$$\left\| M_\varepsilon^{\theta^T} M_\varepsilon^\theta + \lambda \mathbf{I} - \frac{1}{2\varepsilon} \Delta \Delta^T \right\|_F^2 = \operatorname{Tr}\left( \left( M_\varepsilon^{\theta^T} M_\varepsilon^\theta \right)^2 \right) + 2\lambda \operatorname{Tr}\left( M_\varepsilon^{\theta^T} M_\varepsilon^\theta \right) - \frac{1}{\varepsilon} \|M_\varepsilon^\theta \Delta\|_2^2 \tag{22}$$

$$+ \underbrace{\lambda^2 D + \frac{1}{4\varepsilon^2} \|\Delta\|_2^4 - \frac{\lambda}{\varepsilon} \|\Delta\|_2^2}_{\text{independent of } \theta} \tag{23}$$

$$= \|M_\varepsilon^{\theta^T} M_\varepsilon^\theta\|_F^2 + 2\lambda \|M_\varepsilon^\theta\|_F^2 - \frac{1}{\varepsilon} \|M_\varepsilon^\theta \Delta\|_2^2 + C. \tag{24}$$

Hence we get the low rank with Tikhonov regularization loss $\mathcal{L}_{LR}^\lambda = \|M_\varepsilon^{\theta^T} M_\varepsilon^\theta\|_F^2 + 2\lambda \|M_\varepsilon^\theta\|_F^2 - \frac{1}{\varepsilon} \|M_\varepsilon^\theta \Delta\|_2^2$ and setting $\lambda = 0$ yields the low rank loss $\mathcal{L}_{LR} = \|M_\varepsilon^{\theta^T} M_\varepsilon^\theta\|_F^2 - \frac{1}{\varepsilon} \|M_\varepsilon^\theta \Delta\|_2^2$. Note that since $\|M_\varepsilon^{\theta^T} M_\varepsilon^\theta\|_F^2 = \operatorname{Tr}\left( M_\varepsilon^{\theta^T} M_\varepsilon^\theta M_\varepsilon^{\theta^T} M_\varepsilon^\theta \right) = \operatorname{Tr}\left( \left( M_\varepsilon^\theta M_\varepsilon^{\theta^T} \right) \left( M_\varepsilon^\theta M_\varepsilon^{\theta^T} \right) \right)$, and $\|M_\varepsilon^\theta\|_F^2 = \operatorname{Tr}\left( M_\varepsilon^\theta M_\varepsilon^{\theta^T} \right)$, both terms can be computed without ever materializing any $D \times D$ matrices.

## D. Training Algorithms

---

**Algorithm 1** Minibatch training with per-sample noise for Conditional CDC Matching (scalar)

---

**Require:** Dataset $\mathcal{D} = \{x_i\}_{i=1}^N \subset \mathbb{R}^D$; batch size $B$; noise-scale sampler $p(\varepsilon)$; fixed $f, h : \mathbb{R}^D \to \mathbb{R}$; network $\Gamma^\theta : \mathbb{R}^D \times \mathbb{R}_+ \to \mathbb{R}$; optimizer Opt.

1: **for** each training step **do**
2:     Sample minibatch $X = \{X_b\}_{b=1}^B \sim \mathcal{D}$.
3:     For each $b$: sample $\varepsilon_b \sim p(\varepsilon)$.
4:     For each $b$: sample $Y_b \sim \mathcal{N}(X_b, \varepsilon_b \mathbf{I})$.
5:     Compute $T_b \leftarrow \left( f(X_b) - f(Y_b) \right) \left( h(X_b) - h(Y_b) \right) / 2\varepsilon_b$         // conditional CDC target
6:     Predict $P_b \leftarrow \Gamma^\theta(Y_b, \varepsilon_b)$
7:     Loss $\mathcal{L} \leftarrow \frac{1}{B} \sum_{b=1}^B (P_b - T_b)^2$
8:     Update $\theta \leftarrow \operatorname{Opt}(\theta, \nabla_\theta \mathcal{L})$
9: **end for**

---

---

**Algorithm 2** Minibatch training with per-sample noise for Conditional Riemannian Metric Matching (matrix)

---

**Require:** Dataset $\mathcal{D} = \{x_i\}_{i=1}^N \subset \mathbb{R}^D$; batch size $B$; noise-scale sampler $p(\varepsilon)$; network $\Gamma^\theta : \mathbb{R}^D \times \mathbb{R}_+ \to \mathbb{S}^D$ (predicts symmetric $D \times D$); optimizer Opt.
  1: **for** each training step **do**
  2:     Sample minibatch $X = \{X_b\}_{b=1}^B \sim \mathcal{D}$.
  3:     For each $b$: sample $\varepsilon_b \sim p(\varepsilon)$ .
  4:     For each $b$: sample $Y_b \sim \mathcal{N}(X_b, \varepsilon_b \mathbf{I})$.
  5:     Set $\Delta_b \leftarrow X_b - Y_b$.
  6:     Target $T_b \leftarrow (\Delta_b \Delta_b^\top)/2\varepsilon_b \in \mathbb{R}^{D \times D}$                                // conditional metric target
  7:     Predict $P_b \leftarrow \Gamma^\theta(Y_b, \varepsilon_b) \in \mathbb{R}^{D \times D}$
  8:     Loss $\mathcal{L} \leftarrow \dfrac{1}{B} \sum\limits_{b=1}^B \|P_b - T_b\|_F^2$
  9:     Update $\theta \leftarrow \text{Opt}(\theta, \nabla_\theta \mathcal{L})$
 10: **end for**

---

# E. Experimental details

## E.1. Bandwidth sampling

In practice we exprimented with two different noise/bandwidth $\varepsilon$ sampling strategies, namely the uniform noise sampling, and log-normal distribution, following the noise scheduling strategy of Karras et al. (2022). Specifically, for the log-normal strategy we draw $\log \varepsilon \sim \mathcal{N}(P_{\text{mean}}, P_{\text{std}}^2)$ with $P_{\text{mean}} = -1.2$ and $P_{\text{std}} = 1.2$, and clamp the resulting values to the interval $[\varepsilon_{\min}, \varepsilon_{\max}]$. This choice biases sampling toward smaller bandwidths, which are critical for resolving fine-scale geometric structure.

## E.2. Noise level encoding

To condition the network on the bandwidth/noise scale $\varepsilon$, we use Fourier feature embeddings of $t$ of the form $e_\omega(\varepsilon) = (\cos(\varepsilon\omega), \sin(\varepsilon\omega))$. We use $d/2$ frequencies given by $\omega_k = T_{\max}^{-\frac{k}{d/2}}$ for $k = 0, \ldots, \frac{d}{2} - 1$, following standard practice in diffusion and flow-matching models.

## E.3. Synthetic spheres experiment

**FiLM-conditioned residual MLP encoder.** For the synthetic sphere experiments, we parameterize the low-rank factor $M_\varepsilon^\theta(\mathbf{y}) \in \mathbb{R}^{r \times D}$ with a FiLM-conditioned residual MLP (Perez et al., 2018), denoted `MLPEncoder`. The network takes as input a point $x \in \mathbb{R}^D$ and a per-sample noise/bandwidth scalar $\varepsilon$, and outputs

$$\texttt{MLPEncoder}(h, x) \in \mathbb{R}^{r \times D}.$$

This output is used in the low-rank CDC parameterization $\Gamma_\varepsilon^\theta(\mathbf{y}) = M_\varepsilon^\theta(\mathbf{y})^\top M_\varepsilon^\theta(\mathbf{y})$ (or equivalently in the low-rank loss that avoids materializing $\Gamma_\varepsilon^\theta$).

**Output bias.** Optionally, we add an output bias $b_{\text{out}} \in \mathbb{R}^{r \times D}$ to $M_\varepsilon^\theta(\mathbf{y})$ before making $\Gamma_\varepsilon^\theta$, which we find help during early training.

**Noise-scale conditioning.** We condition the network on $\varepsilon$ using a sinusoidal noise-level embedding as describe in Sec. E.2 followed by a small MLP. Concretely, we compute feed the noise-level encoding in a 2-layer MLP with SiLU nonlinearity. This embedding is injected into *every* residual block via FiLM (feature-wise linear modulation) (Perez et al., 2018), rather than concatenated only once at the input.

**Initialization.** We initialize each residual block near the identity map, meaning a zero initialization of the projection heads of each blocks. Importantly, we do *not* zero-initialize the final projection head: $W_{\text{out}}$ is initialized from a zero-mean Gaussian with small standard deviation, and $b_{\text{out}} = 0$. This avoids an initial output $M_\varepsilon^\theta(\mathbf{y}) \approx 0$, which can lead to weak gradients when optimizing losses involving $M^\top M$.

**Synthetic sphere configuration.** Unless otherwise specified, we use ambient dimension $D = 64$ and sphere dimension $d = 8$. The architecture used hidden width $H = 1024$, $L = 4$ residual blocks, rank $r = 16$, and time/noise embedding enabled. We train with AdamW (learning rate $10^{-4}$, no weight decay) for 3000 epochs with batch size 1024 and gradient clipping with norm 1. In this configuration the MLP has roughly 15M parameters. The noise level sampling strategy used during training was log-normal with $h_{\min} = 10^{-4}$ and $h_{\max} = 16$.

**Evaluation and hyper-parameter tuning for Fig. 2.** We select evaluation hyper-parameters via grid search on a held-out validation set, independently for each training-set size. For metric matching, the only tuned hyper-parameter is the conditioning bandwidth $\varepsilon$; we evaluate $\varepsilon = 2^{-k}$ for $k = -9, -8, \ldots, 4$ and report test performance using the value that minimizes the validation error. For the $k$-NN graph baseline, we tune both the number of neighbors and the bandwidth, searching over $k = 2^p$ $p = 3, 4 \ldots, 11$ and $\varepsilon = 2^{-k}$ for $k = -9, -8, \ldots, 4$, and again report test performance for the configuration achieving the lowest validation error.

## E.4. MNIST experiments

**UNet backbone.** We use a standard timestep-conditioned UNet backbone with residual blocks and attention from (Dhariwal & Nichol, 2021). After the final UNet layer, the network outputs a tensor of shape $[B, K, W, H]$, where $B$ is the batch size, $W$ and $H$ are spatial dimensions, and $K$ is the number of output channels. We then reshape this output to $[B, K, WH]$ (or equivalently flatten the spatial dimensions) and interpret the channel dimension as a collection of $K$ per-pixel feature maps. In our metric-matching parameterization, these $K$ channels represent the entries of a low-rank factor: we set $K = Cr$ where $C$ is the input channels (so ambient dimension $D = CWH$) and reshape the flattened output into a matrix $U_\theta(x) \in \mathbb{R}^{D \times r}$ (per input), where $D$ is the ambient dimension of the data and $r$ is the chosen rank. We then form the predicted (co)metric as the low-rank PSD matrix

$$\hat{\Gamma}_\theta(x) = U_\theta(x) U_\theta(x)^\top \in \mathbb{R}^{D \times D}.$$

Finally, we optionally add a learned output bias at the UNet head: after the last convolution and before reshaping into $U_\theta(x)$, we add a trainable vector $b \in \mathbb{R}^K$ to the $K$ output channels and broadcast it across spatial dimensions, followed by the reshaping described above.

**MNIST configuration.** For MNIST we use $28 \times 28$ inputs with `model_channels`$= 64$, `num_res_blocks`$= 2$, `channel_mult`$(1, 2, 2)$, and enable attention at downsampling factor 4 (i.e., $7 \times 7$ feature maps). We additionally include a learnable output bias term which we initialize a centered normal distribution with $1e - 3$ variance, as it empirically improves optimization for early stages of training. We use an output rank of $r = 100$ for the low rank parameterization, and use low rank training. We use Adam optimizer with learning rate $2 \times 10^{-4}$, $\beta_1 = 0.9$, $\beta_2 = 0.999$, and $\epsilon = 10^{-8}$, and no weight decay. Training is run for up to 1500 epochs with batch size 128. We apply gradient clipping with global $\ell_2$-norm threshold 1.0. Each training step samples a noise/scale parameter $h \in [h_{\min}, h_{\max}]$ with $h_{\min} = 10^{-4}$ and $h_{\max} = 25$ using a uniform sampling scheme. We trained using gaussian augmentation with 0.1 standard deviation.

**Inference parameters.** We use a bandwidth of $\varepsilon = 7$ as conditioning bandwidth in order to generate Fig. 3a, Fig. 3b and Fig. 5. For the Riemannian optimization experiment we Riemannian gradient descent with constant learning rate 0.01 and $2^{13}$ steps. Additionally, we add a slight isotropic component to the metric replacing $\Gamma^\theta$ by $(1 - \delta)\Gamma^\theta + \delta\mathbf{I}$ with $\delta = 10^{-4}$ in order to allow for small movement in all direction, as it improved convergence of the algorithm.

**CelebA configuration.** For CelebA we use $64 \times 64$ RGB inputs with `model_channels`$= 64$, `num_res_blocks`$= 3$, `channel_mult`$(1, 2, 2, 2)$, and enable attention at downsampling factors 4 and 8. We include a learnable output bias term and use an output rank of $r = 128$ for the low-rank parameterization and use low rank training together with the mean-centered loss. Each training step samples a noise/scale parameter $h \in [h_{\min}, h_{\max}]$ with $h_{\min} = 1.0$ and $h_{\max} = 5.0$ using a uniform sampling scheme. We use the AdamW optimizer with learning rate $5 \times 10^{-5}$, $\beta_1 = 0.9$, $\beta_2 = 0.999$, and $\epsilon = 10^{-8}$, with no weight decay. Training is run for up to 200 epochs with batch size 256. We apply gradient clipping with threshold 1.0.

For the CelebA posterior mean model used for mean-centered Riemannian metric matching, we use $64 \times 64$ RGB inputs with `model_channels`$= 128$, `num_res_blocks`$= 3$, `channel_mult`$(1, 2, 2, 2)$, and enable attention at downsampling factors 4 and 8. Each training step samples a noise/scale parameter $h \in [h_{\min}, h_{\max}]$, with $h_{\min} = 0.002$ and $h_{\max} = 20.0$, using a uniform sampling scheme. We use the Adam optimizer with learning rate $2 \times 10^{-4}$, $\beta_1 = 0.9$, $\beta_2 = 0.999$, and $\epsilon = 10^{-8}$, with no weight decay. Training is run for 100 epochs. We apply gradient clipping with threshold 1.0.

**CelebA configuration.** For FFHQ, we use $256 \times 256$ RGB inputs with `model_channels`$= 128$, `num_res_blocks`$= 2$, `channel_mult`$(1, 2, 2, 4)$, and enable attention at downsampling factors 8 and 16. We include a learnable output bias term and use an output rank of $r = 256$ for the low-rank parameterization, together with low-rank training and the mean-centered loss. Each training step samples a noise/scale parameter $h \in [h_{\min}, h_{\max}]$, with $h_{\min} = 1.0$ and $h_{\max} = 5.0$, using a uniform sampling scheme. We use the AdamW optimizer with learning rate $5 \times 10^{-5}$, $\beta_1 = 0.9$, $\beta_2 = 0.999$, and $\epsilon = 10^{-8}$, with no weight decay. Training is run for up to 80 epochs with batch size 32 using `bf16-mixed` precision. We apply gradient clipping with threshold 1.0.

For the FFHQ posterior mean model, we use $256 \times 256$ RGB inputs with `model_channels`$= 128$, `num_res_blocks`$= 2$, `channel_mult`$(1, 2, 2, 4)$, and enable attention at downsampling factors 8 and 16. Each training step samples a noise/scale parameter $h \in [h_{\min}, h_{\max}]$, with $h_{\min} = 0.002$ and $h_{\max} = 20.0$, using a uniform sampling scheme. We use the Adam optimizer with learning rate $2 \times 10^{-4}$, $\beta_1 = 0.9$, $\beta_2 = 0.999$, and $\epsilon = 10^{-8}$, with no weight decay. Training is run for 200 epochs with `bf16-mixed` precision. We apply gradient clipping with threshold 1.0.

# F. Additional results

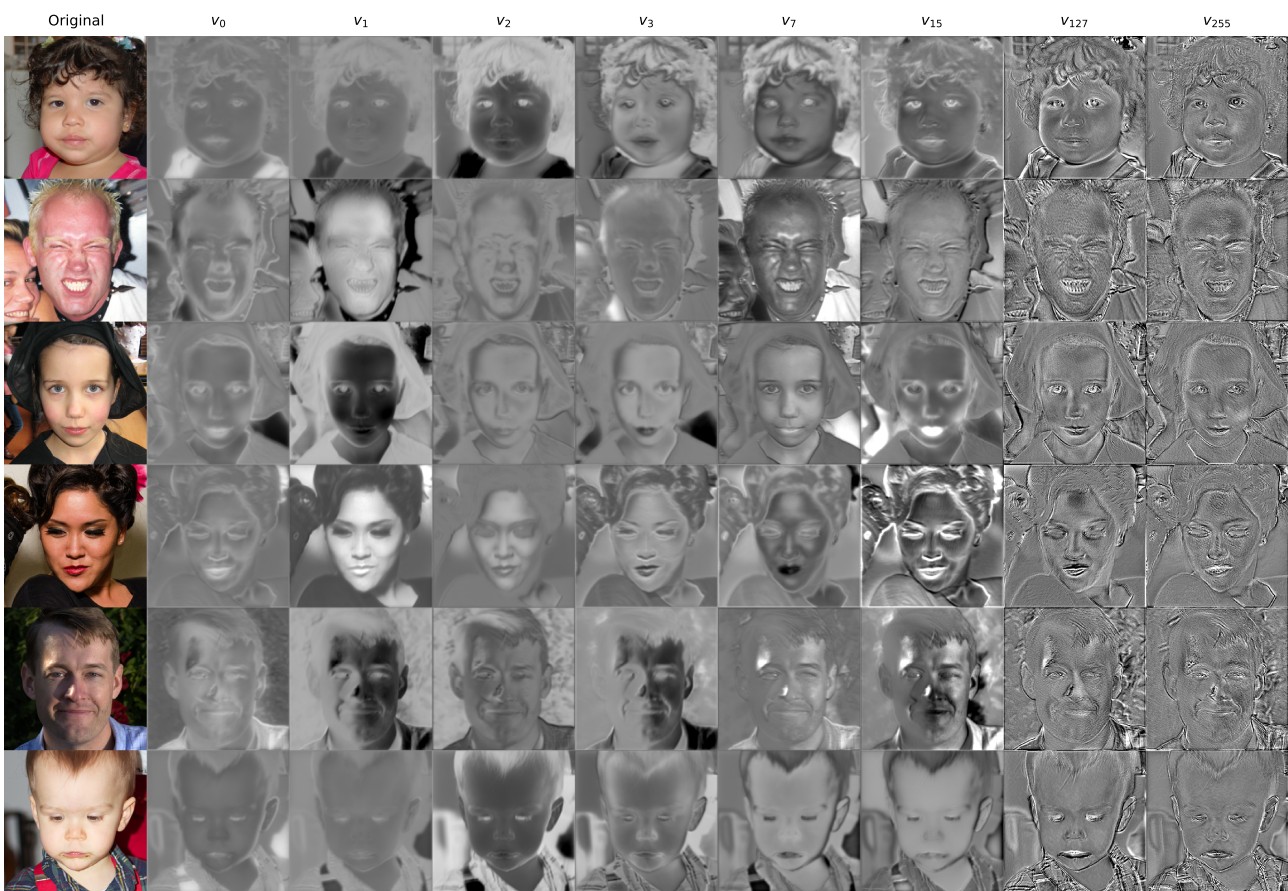

*Figure 6.* Visualization of the first 4, 8th, 16th, 128th, and 256th eigenvectors of the CDC matrix learned by metric matching on FFHQ. The tangent vectors are shown in grayscale for greater clarity.

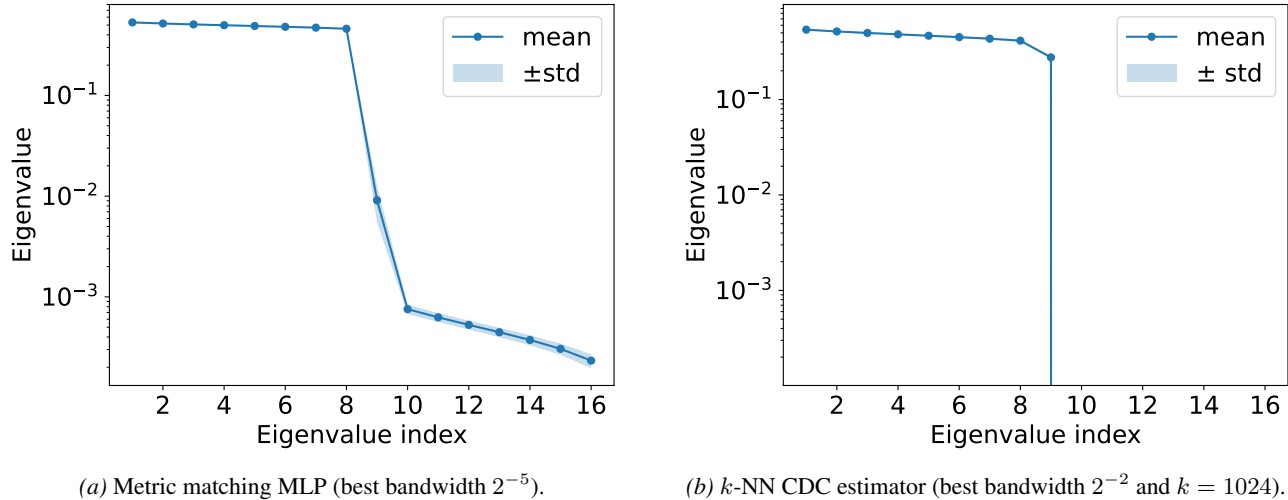

*(a)* Metric matching MLP (best bandwidth $2^{-5}$).  *(b)* $k$-NN CDC estimator (best bandwidth $2^{-2}$ and $k = 1024$).

*Figure 7.* Mean ordered top 16 eigenvalues on the 8-dimensional sphere (512k samples), for the best-performing tangent space predictor hyperparameters of (left) metric matching and (right) a $k$NN-based CDC estimator.

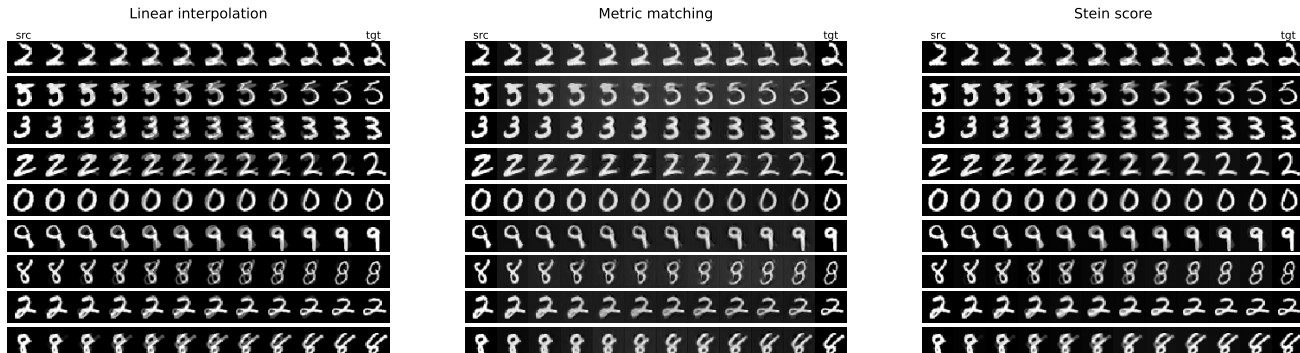

*Figure 8.* Interpolation on MNIST. We compare linear interpolation (LERP), metric matching, and the Stein score metric on the interpolation problem.

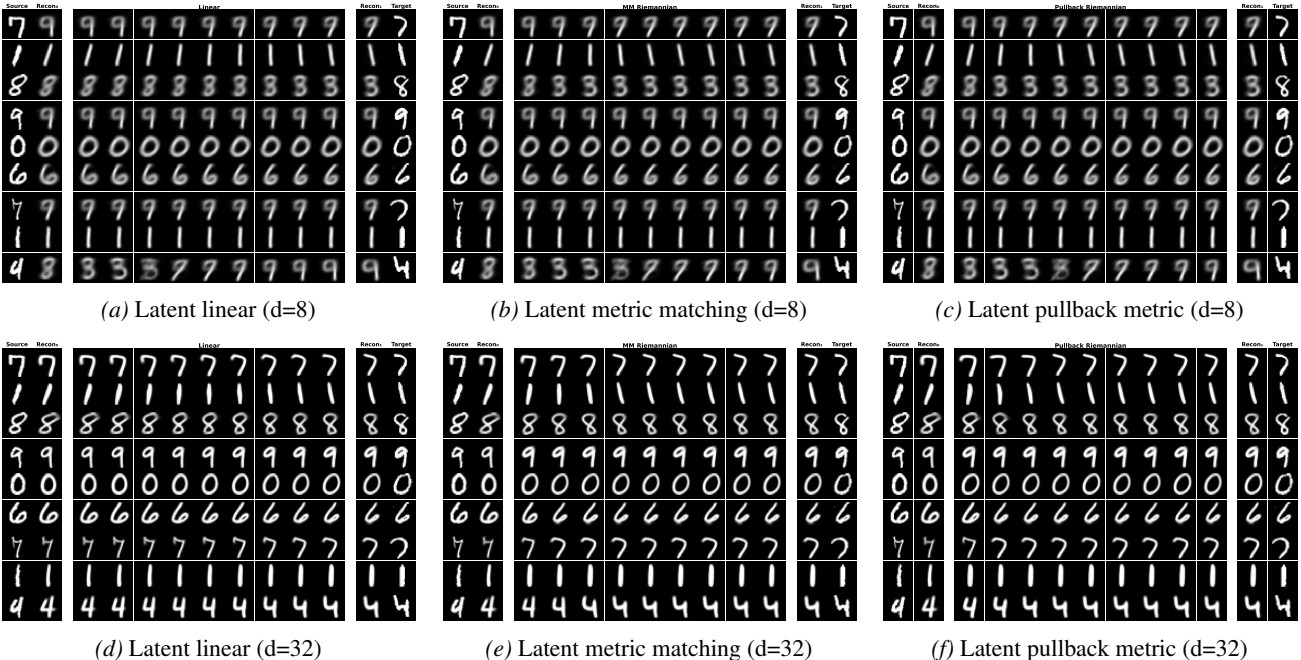

*(a)* Latent linear (d=8)  *(b)* Latent metric matching (d=8)  *(c)* Latent pullback metric (d=8)

*(d)* Latent linear (d=32)  *(e)* Latent metric matching (d=32)  *(f)* Latent pullback metric (d=32)

*Figure 9.* Interpolation on MNIST in latent space of a VAE. Top row shows the smaller VAE with 2M parameters and latent dimension $d = 8$; bottom row shows the larger VAE with 12M parameters and $d = 32$. We compare linear interpolation, metric matching, and the pullback metric on the interpolation optimization problem.

