# OpenReview forum: "Riemannian Metric Matching for Scalable Geometric Modeling of Distributions"
_ICML.cc/2026/Conference — ICML 2026 spotlight_

### Official Review · Reviewer_Bh9d · 2026-03-04

**Soundness:** 3
**Presentation:** 3
**Significance:** 3
**Originality:** 3
**Overall Recommendation:** 5
**Confidence:** 3

**Summary:**

It is an open question as for how to efficiently recover Riemannian structure from high dimensional datasets. The paper proposes a novel objective for learning the carré du champ operator from data using an explicit objective through the marginalization trick. Empirical evidence shows that the proposed method is capable of learning reasonable Riemannian structures from datasets while being highly scalable.

**Compliance With Llm Reviewing Policy:**

Affirmed.

**Final Justification:**

The authors provided further empirical evidence, addressing my questions, and I maintain my original positive rating.

**Key Questions For Authors:**

In terms of experiments, the largest scale experiments seem to be on MNIST, which is still a relatively simple dataset. Have the authors tried scaling the proposed method to larger datasets?

**Limitations:**

yes

**Strengths And Weaknesses:**

Strengths:

1. The proposed approach provides a scalable approach to learn the Riemannian structure with theoretical guarantees.

2. Empirical evidence considers multiple applications, demonstrating the benefits of the learned Riemannian structure.

Weaknesses:

1. I find the paper fails to discuss some related works; e.g. [1] also tackles the problem of learning geometry from data using neural networks, and (not strictly relevant) [2] presented an alternative perspective on the geometry in diffusion models.

[1] Neural FIM for learning Fisher information metrics from point cloud data, Fasina et al., ICML 2023

[2] The Spacetime of Diffusion Models: An Information Geometry Perspective, Karczewski et al., ICLR 2026

---

> ### Author Rebuttal · Authors · 2026-03-31
>
> We thank the reviewer for the encouraging feedback. We are pleased that the reviewer found the approach novel and highly scalable, and that our work demonstrates the benefits of the learned Riemannian structure. We now address the weaknesses and questions in full.  To support the replies, we have attached additional figures in an anonymous link: https://anonymous.4open.science/r/icml2026_anonymous-058C/rebuttal_figures.pdf.
>
> **Additional related works.** We thank the reviewer for pointing out these important references. We will add them in the related works section.
>
> **Have the authors tried scaling the proposed method to larger datasets?** We thank the reviewer for the suggestion. We have indeed scaled the proposed method to larger datasets, namely cifar-10 (60k colored 32x32 images), celebA (200k colored 64x64), and FFHQ (70k colored 256x256), which are considerably larger than MNIST. We report results in the linked figures, and detail what we report below, for other reviewers to see.
>
> **Additional experiments.**
> Following reviewer feedback to more thoroughly evaluate the scalability of metric matching, we have evaluated it on three additional datasets: CIFAR-10 (60k colored 32x32 images, D = 3072), celebA (200k colored 64x64, D=12288), and FFHQ (70k colored 256x256, D=196608), which are considerably larger than MNIST (D=728). These examples lack a ground truth for evaluating and comparing models, so we introduce the following Tangent Inception Distance (TID) metric. We compute features from a standard pretrained inception network used for FID in generative modelling (clean-fid implementation), and consider how those features vary when the input image is perturbed. When the perturbation is tangent to the manifold, the difference in features should be small (see e.g. [1]), so we define TID to be the average perturbation distance. This serves as a metric for the quality of tangent space estimates where no ground truth exists. Given an estimated tangent space in the form of a covariance matrix, we can sample perturbations and measure the distance. As a baseline, we add perturbation in random directions, which corresponds to the Euclidean (isotropic) metric.
>
> We can use this evaluation to benchmark additional methods against ours. Following reviewer suggestions, we have implemented the metric based on the Jacobian of the score, which has been used to estimate tangent spaces in [2], and the Stein score-based metric introduced in [3]. The Stein score metric is a rough approximation given by a rank-1 correction to the ambient identity matrix, whose effect is pronounced in lower dimensions but vanishes as dimension increases (we see a slight improvement on randomness for MNIST and CIFAR but virtually no gain on CelebA and FFHQ). The score Jacobian is also a principled estimator for the true geometry [2] and performs much better, but underperforms metric matching. We expect that this is because it relies on implicit information extracted from the score, whereas metric matching is trained explicitly as a metric. An additional practical constraint is the high computational cost of computing the Jacobian, which requires **~3000x more memory** and is **~62x slower** per evaluation on CIFAR-10, and results in OOM on CelebA and FFHQ (despite 80GB GPU memory), whereas the Stein score metric and our method scale easily to high dimensions.
>
> We thank the reviewer for their time and valuable feedback. We welcome the opportunity to further discuss any remaining concerns.
>
>
> ___________________________________
> [1] Srinivas, S., Bordt, S., & Lakkaraju, H. (2023). Which models have perceptually-aligned gradients? an explanation via off-manifold robustness. Advances in neural information processing systems, 36, 21172-21195.
>
> [2] Kharitenko et al., "Landing with the score: Riemannian optimization through denoising." ICLR, 2026
>
> [3] Azeglio et al., "What's inside your diffusion model? A score-based Riemannian metric to explore the data manifold." ArXiv, 2025

---

> > ### Author Rebuttal · Reviewer_Bh9d · 2026-04-03
> >
> > I thank the authors for their response. I maintain my original positive rating.

---

> > > ### Author Response · Authors · 2026-04-08
> > >
> > > We thank the reviewer for their time and positive review of our work.

---

### Official Review · Reviewer_qeQP · 2026-03-08

**Soundness:** 3
**Presentation:** 3
**Significance:** 2
**Originality:** 3
**Overall Recommendation:** 5
**Confidence:** 3

**Summary:**

This paper proposes Riemannian metric matching to learn the geometry of data distributions. The main idea is to learn the carre du champ (CDC) operator by reformulating it as a conditional expectation over random perturbations, therefore allows training-based convergence without the need for kNN constructions.

**Compliance With Llm Reviewing Policy:**

Affirmed.

**Final Justification:**

This is an interesting paper, good read and the introduction of CDC for this application is neat. Authors sufficiently addressed my review concerns as well as follow-up questions after the rebuttal. I therefore updated my overall score from 4 to 5.

**Key Questions For Authors:**

1. When can we assume a dataset has smooth density? Or even better, what would be the way for someone to "induce" smooth density on a true real-world data distribution?
2. Can the authors elaborate on their claim "When Riemannian metric is different ... paths are not always guaranteed to converge to $\textbf{p}$ ... but we expect this to pose less of a problem in high dimensions". How can we assume this is less impactful at high dimensions? This claim needs either formal justification or empirical evidence.
3. Can the authors compare both synthetic spheres benchmarks and synth data generation (interpolation along data manifold) with recent published methods (such as [2] for spheres and [4, 5] for interpolation)? For the interpolation task, quantitative metrics such as FID scores against methods are necessary.

**Strengths And Weaknesses:**

**Strengths:**
- Estimating underlying Riemannian geometry without the need for graph-based constructions is well motivated.
- The use of the carre du champ operator for this is novel (to the best of my knowledge).
- A good portion of the theoretical portions connecting riemannian metrics, diffusion geometry and carre du champ are well explained and interesting. Although some assumptions are constricting from a real-world data perspective (see weaknesses).

**Weaknesses:**
- The conditional CDC requires a smooth density of a true data distribution. This seems difficult to hold for real datasets. This is different from the smooth functions or manifolds which we standard. The densities being smooth, though, I think is difficult to assume due to the potential sparsity of observed samples [1]
- No experimental comparisons to previous literature, such as general diffusion geometry methods. For example, synthetic speedup is compared only between neural surrogate-based CDC and kNN-based CDC. Current diffusion geometry methods [2] use the Hessian as a score function from diffusion models to perform Riemannian optimization or [3] which exploits scores at small noise levels for intrinsic dimension estimation. Without these comparisons, its difficult to contextualize the contribution relative to current methods.
- Experimental sections are a bit weak. Only looking at synthetic spheres and MNIST (for real datasets). I would like to see a higher number of dataset baselines (such as including CIFAR) as well as exploration into other domains (perhaps exploring different manifolds such as torus, etc.)?
- Synthetic data generation is only compared to LERP (simple regression-based) and does not explore the fidelity of manifold interpolation to more recent literature such as [4, 5].

---
**References:** \
[1] Berry et al., "Density estimation on manifolds with boundary." _Computational Statistics & Data Analysis_, 2017 \
[2] Kharitenko et al., "Landing with the score: Riemannian optimization through denoising." _ICLR_, 2026 \
[3] Stanczuk et al., "Diffusion models encode the intrinsic dimension of data manifolds." _ICML_, 2024 \
[4] Azeglio et al., "What's inside your diffusion model? A score-based Riemannian metric to explore the data manifold." _ArXiv_, 2025 \
[5] Kapusniak et al., "Metric flow matching for smooth interpolations on the data manifold." _NeurIPS_, 2024

---

> ### Author Rebuttal · Authors · 2026-03-31
>
> We thank the reviewer for the thoughtful and constructive feedback. We appreciate the positive assessment of the motivations, the novelty of the carré du champ formulation, and the clarity of the theoretical development. We address the concerns below.  To support the replies, we have attached additional figures in an anonymous link: https://anonymous.4open.science/r/icml2026_anonymous-058C/rebuttal_figures.pdf.
>
> **(W3) Additional datasets.**
> We agree that the experimental section can be strengthened. Following the reviewer’s suggestion, we have added CIFAR, CelebA, and FFHQ, covering substantially higher-dimensional and more realistic datasets. We first visualize the learned geometry in Figure 1, where we visualize the eigenvectors of the learned metric in grayscale. To evaluate the predicted tangent directions, introduce a Tangent Inception Distance metric: we perturb images along learned tangent directions and compare the induced change in a pretrained Inception feature space (Figure 2), and expect good tangent directions should minimally affect features. We report results in Figure 2, and additionally benchmark metric matching against random perturbations, the score-Jacobian metric from [2], and the Stein score metric for [4] on MNIST, CIFAR-10, CelebA, and FFHQ, and report both performance in Figure 2 and scaling (runtime/memory) in Table 1. These results show that the learned tangent directions capture meaningful semantic structure, and clearly outperform the random ambient perturbation as well as the baselines. We have detailed these experiments in full in our response to reviewer Bh9d.
>
> **(W2/Q3) Comparison to recent methods.**
> We thank the reviewers for highlighting [2] as an important baseline. We have now run this comparison on MNIST and on CIFAR-10. On the higher dimensional datasets, however, we were not able to run the baselines since computing the Jacobian causes out of memory error despite the 80GB memory. On CIFAR-10, the score Jacobian requires ~3000x more memory and is ~62x slower per evaluation. We observe that in the perturbation-based evaluation described above, our method tends to produce more stable perturbations, as measured by a lower average inception distance change on both MNIST and CIFAR-10. While producing higher-quality perturbations, our method is also more scalable due to the amortized forward pass inference, making it more practical in high-dimensional regimes. We will include this comparison in the revision.
>
> **(W4/Q3) Interpolation comparison.** We thank the reviewer for this suggestion. We have added a comparison to [4] by adding their proposed metric into our interpolation pipeline, enabling a controlled comparison where only the metrics differ. Our metrics yield more stable and semantically meaningful interpolation trajectories. This is due to the fact that the metric used in [4] is a rank-one perturbation of the standard Euclidean distance, and therefore the geometry is close to Euclidean: see Figure 3.
>
> We have also added this method in the perturbation experiment on high-dimensional datasets, which shows behaviour almost identical to the random perturbation, confirming the similarity to the Euclidean ambient geometry. In particular, since this metric only differs from the Euclidean metric in one dimension, we see that the gap from it to random perturbations vanishes as the dimension increases. For [5], the method involves training an additional model on top of the metric, making a direct controlled comparison less straightforward. We will add these comparisons and clarify the distinctions in the revision.
>
> **(W1/Q1)  Assumptions on the density.** The density assumptions in Theorem 3.1 are standard in the flow/score matching literature (and can apply with a general measure, let alone a smooth density), while those in Theorem 4.1 follow classical diffusion maps results. We agree that real-world data is only observed through sparse samples in high dimensions. Our perspective is that the smooth density assumption provides a standard theoretical justification, while in practice, we believe that the inductive bias offered by neural networks helps to learn meaningful structures from finite samples.
>
> **(Q2) High-dimension claims.** We thank the reviewer for pointing this out. We agree that the second part of the statement lacks evidence, and have removed it from the text.
>
> We hope that these additional experiments and clarifications address the reviewer’s concerns and strengthen the empirical support for our approach. We would greatly appreciate it if the reviewer could take these updates into account in their final assessment.
>
> ______
>
> [2] Kharitenko et al., "Landing with the score: Riemannian optimization through denoising." ICLR, 2026
>
> [4] Azeglio et al., "What's inside your diffusion model? A score-based Riemannian metric to explore the data manifold." ArXiv, 2025
>
> [5] Kapusniak et al., "Metric flow matching for smooth interpolations on the data manifold." NeurIPS, 2024

---

> > ### Author Rebuttal · Reviewer_qeQP · 2026-04-03
> >
> > I thank the authors for their detailed rebuttal and substantial additional experiments. The rebuttal addresses several of my original concerns meaningfully. The scaling results showing the score-Jacobian method running OOM on CelebA and FFHQ (despite 80GB GPU memory) are interesting practical observation that strengthens the paper's motivation. I do not penalize the authors for the missing baselines on those two datasets, as this is itself a useful empirical finding.
> >
> > However, I have follow-up concerns regarding the Tangent Inception Distance (TID) metric:
> > 1. TID is conceptually very close to Perceptual Path Length (PPL) from [6], which measures perceptual feature change under small latent perturbations. TID applies the same logic (perturbation along a good direction should produce minimal change in a pretrained feature space) but in pixel space with Inception features rather than in latent space with LPIPS. Given this close relationship, can you discuss how TID relates to PPL, and whether known limitations of PPL (sensitivity to step size, feature extractor choice) also apply?
> > 2. More critically, TID is introduced without independent validation. On the synthetic sphere experiments where ground-truth tangent spaces are available, you use Frobenius distance. Can you show that TID correlates with Frobenius error in that controlled setting? This would build confidence that TID is a reliable proxy for tangent space quality when ground truth is unavailable.
> > 3. Is TID controlled for perturbation magnitude? A method could achieve low TID simply by producing very small perturbations regardless of their geometric correctness. Clarification on normalization would be helpful.
> > 4. On CelebA and FFHQ, the method is compared only against random perturbations and the Stein score metric (which you note in your response to Bh9d, behaves nearly identically to random in high dimensions). While the Jacobian OOM issue is understandable, this still means the method is effectively uncontested on the highest-dimensional datasets. Are there any additional scalable baselines that could be included?
> >
> > I appreciate the effort in the rebuttal and am willing to revisit my score after response to these additional questions. As of now, I maintain my current score of 4 for now.
> >
> > ---
> > [6] Karras et al., "A style-based generator architecture for generative adversarial networks." _CVPR_, 2019
> >
> > ---
> > ### Edit after Reply Rebuttal Comment by Authors
> > The discussion on points related to TID are helpful and engaging. My questions are sufficiently answered and update my score to 5.

---

> > > ### Author Response · Authors · 2026-04-06
> > >
> > > 1. We thank the reviewer for noting the relationship between TID and PPL, which we had not considered. Whereas PPL measures the overall quality of a path through the space, TID measures the quality of a *local* perturbation. One can think of TID as a one-step PPL over a very short straight-line path, rather than a multi-step sum over a long path. We compute TID over a range of perturbation sizes $\alpha$ (see discussion in Q3 below), but are really interested in the local quality of the perturbation, i.e. the slope of the TID curve as $\alpha\to0$. By plotting a slope over a range of $\alpha$ values, we remove this dependency as felt by PPL (but at the expense of producing a curve rather than a single number). The TID shares the same feature-extractor dependence as PPL, and we would like to strengthen this metric in future by averaging over multiple choices. We will discuss this relationship in the final draft of the paper.
> > >
> > > 2. This is a really interesting question and points to a major challenge in evaluating geometric methods: namely, that there are few examples with both a formal ‘ground truth’ and a deep learning proxy metric like TID, PPL, FID etc. In the case of synthetic manifolds, we can evaluate the ground truth Frobenius norm, but do not have a relevant feature-extractor network to compute TID. We could use a synthetic proxy feature, like distance to the manifold, but this is quite artificial and not really comparable to a feature from a deep network. We would be very interested to hear of any examples where both a meaningful inception network and ground truth tangent spaces exist.
> > >
> > > 3. The TID is explicitly controlled for perturbation magnitude by normalising the length to an explicit $\alpha$, for exactly this reason. As such, the perturbations from the unit sphere around each point, with density given by the metric (interpreted as a covariance). This is important to fairly compare models of different ranks, because low-rank models would generally attain better scores if left unnormalized. Since we train metric matching with a low-rank model, the unnormalized results would be unfair on the other full-rank benchmarks. We will clarify this in the final draft of the paper.
> > >
> > > 4. We feel that this issue exactly highlights the challenges of scaling geometric methods to high dimensions. Since the Riemannian metric is represented by a $D \times D$ matrix, it seems impossible to scale to large $D$ without using a low-rank factorization. Here, we use metric matching with a rank-$k$ factorization, with a guarantee that one can take $k = 2d-1$ (where $d$ is the intrinsic dimension), leading to an $\mathcal{O}(dD)$ complexity instead of $\mathcal{O}(D^2)$. We see this in Table 1, where the memory of metric matching scales linearly, while the score Jacobian method scales quadratically. While the score Jacobian has similar theoretical guarantees and is impressive for low $D$, there is no way to reduce its rank because it is obtained indirectly from a model trained with score matching. Conversely, metric matching explicitly learns a metric, with theoretical guarantees, and can be used to train low-rank models. To the best of our knowledge, there are no other models that combine this scalability with the theoretical guarantee to actually approximate the true metric.
> > >
> > > We thank the reviewer again for their detailed and encouraging review and helpful suggestions.

---

### Official Review · Reviewer_n8HN · 2026-03-12

**Soundness:** 3
**Presentation:** 3
**Significance:** 3
**Originality:** 3
**Overall Recommendation:** 4
**Confidence:** 4

**Summary:**

This paper proposes Riemannian metric matching, a denoising-based framework for learning the intrinsic geometry of data by training a neural network to estimate the carré du champ (CDC) operator at each point. The key insight is that the intractable marginal CDC can be rewritten as a conditional expectation over random perturbations, yielding a sample-wise training objective that avoids explicit kernel construction. The authors prove that the learned CDC converges to the true tangent space projector as $\epsilon \rightarrow 0$ in the population limit, and demonstrate empirically that the method matches or outperforms kNN-based estimators on synthetic manifolds while scaling to high-dimensional image data where classical graph-based methods break down.

**Compliance With Llm Reviewing Policy:**

Affirmed.

**Final Justification:**

The additional experiments and analyses appeared during the discussion period adequately address my main concerns. I strongly encourage the authors to incorporate these results into the main paper in the revised version. I am willing to raise my score accordingly.

**Key Questions For Authors:**

- The paper positions itself against latent space-based methods by arguing that Jacobian computation is expensive and lacks theoretical guarantees. However, prior works [1, 2] have demonstrated comparable geometric applications on MNIST and beyond. Could the authors provide a direct comparison with these latent space methods on MNIST, particularly in terms of computational cost and the quality of recovered geometric quantities?
- Could the authors demonstrate the method on substantially higher-dimensional datasets, such as FFHQ or similar high-resolution image data?

## References

[1] Arvanitidis, G., González-Duque, M., Pouplin, A., Kalatzis, D., & Hauberg, S. (2021). Pulling back information geometry. arXiv preprint arXiv:2106.05367.
[2] Arvanitidis, G., Hauberg, S., & Schölkopf, B. (2020). Geometrically enriched latent spaces. arXiv preprint arXiv:2008.00565.

**Limitations:**

The paper lacks an explicit discussion of its limitations and, more critically, omits comparisons against latent space-based Riemannian metric methods, which serve as natural baselines for the demonstrated applications.

**Strengths And Weaknesses:**

## Strengths

- The paper is well-written and easy to follow. The progression from background on diffusion geometry to the proposed conditional metric matching objective is clearly motivated.
- The core idea of approximating the carré du champ operator via a continuous, denoising-style neural surrogate is novel. The reformulation of the marginal CDC as a conditional expectationis elegant.
- The scalability gains over kNN-based CDC estimators are impressive.

## Weaknesses

- **Missing related work.** The paper does not discuss relevant lines of work on learning Riemannian geometry from data [1,2]. These omissions make it difficult to contextualize the contribution relative to the broader landscape of neural Riemannian metric estimation.
- **Limited experimental scope for high-dimensional claims.** The paper's central claim is that metric matching scales to high-dimensional data where kNN methods fail, which is supported by the experiments on MNIST. However, prior works [1, 2] have already demonstrated comparable downstream applications, such as on-manifold interpolation, on both MNIST and CelebA. To substantiate the claimed advantage in high-dimensional settings, experiments on more challenging, higher-resolution image datasets (or other high-dimensional modalities) would be necessary.

---

> ### Author Rebuttal · Authors · 2026-03-31
>
> We thank the reviewer for the valuable and insightful feedback. We are grateful that the reviewer found the method clearly motivated and novel, the marginalization trick elegant, and the scalability gains over previous CDC estimators impressive. We now respond to the weaknesses and questions raised. To support the replies, we have attached additional figures in an anonymous link: https://anonymous.4open.science/r/icml2026_anonymous-058C/rebuttal_figures.pdf.
>
> **(W1, Q1)  Latent space methods.** We thank the reviewer for highlighting [1, 2] and agree that latent-space approaches are an important line of work, and we will include a detailed discussion in the revised version. Our setting differ in that our goal is to estimate the ambient-space metric directly in $R^D$, whereas [1, 2] define pullback metrics in a learned latent space via a learned generator  $f: Z \to R^D$. Concretely, latent space methods introduce 1. an additional generator, 2. make assumptions on the latent dimension and the topology via $Z$ (typically taken to be $R^d$) and 3. require the Jacobian of $f$ to define the pullback (see Q1 below for the computational cost of Jacobian-based methods). In contrast, metric matching requires not additional latent space, make no assumptions on the intrinsic dimension or topology, and is significantly improves computations efficiency (~3000× less memory and is ~62× faster per evaluation on CIFAR-10).
>
> A direct comparison with [1,2] would be to apply metric matching directly in the latent space. This would follow the ideas of latent diffusion models, where the denoising objectives of score and flow-based models have been very successful, and we would hope to get similarly good results there. This would allow a direct evaluation of our method against the naturally latent methods of [1,2]. However, a proper evaluation of metric matching in latent space is beyond the scope of this paper, and we plan to study this topic in full detail, in the context of [1,2], in subsequent work.
>
> In the setting of ambient-space metrics, we have now added benchmarks against two baselines: the score Jacobian-based estimator and the Stein-score metric, as the closest meaningful baselines (see W2 and Q1 below).
>
> We thank the reviewer for this question, and will update the text to make this distinction clearer in the updated text. If the reviewer has a specific comparison setting in mind that would still be meaningful, we would be happy to include it.
>
> **(W2) Higher dimensional datasets.** We agree that stronger high-dimensional validation is important. We have therefore added experiments on CIFAR-10 (D = 3072), CelebA (D=12288), and FFHQ (D=196608), which are considerably larger than MNIST (D=728). To evaluate geometric quality in these settings and benchmark against other metrics, we introduce a Tangent Inception Distance metric: we perturb images along learned tangent directions and compare the induced change in a pretrained Inception feature space (Figure 2). We observe that features are significantly more stable under tangent perturbations than under random perturbations of equal magnitude, indicating that the learned tangents capture semantically meaningful directions. We additionally visualize tangent directions on FFHQ (Figure 1).
>
> We also add two additional baselines: the score Jacobian [1] and Stein score metric [2], and compare the TID curves. We find much better performance from metric matching in each case (wherever the score Jacobian is not inhibited by memory requirements). These experiments further support our claim that the method remains effective in regimes where graph-based methods become impractical. We have detailed these experiments in full in our response to reviewer Bh9d.
>
> **(Q1) Computational cost of Jacobian.** In addition to the discussion above, we include a direct comparison to the score-Jacobian and Stein score baselines, reported in Table 1. On CIFAR-10, Jacobian computation requires *~3000× more memory* and is *~62× slower* per evaluation, and becomes infeasible on CelebA and FFHQ (OOM on H100 80GB GPU), whereas our method scales to these settings.
>
> We thank the reviewer again for the helpful suggestions. We hope that the added experiments and clarified positioning address the main concerns regarding evaluation and related work, and we would appreciate a reevaluation of our work. We welcome the opportunity to further discuss any remaining concerns.
>
> [1] Kharitenko et al., "Landing with the score: Riemannian optimization through denoising." ICLR, 2026
>
> [2] Azeglio et al., "What's inside your diffusion model? A score-based Riemannian metric to explore the data manifold." ArXiv, 2025

---

> > ### Author Rebuttal · Reviewer_n8HN · 2026-04-03
> >
> > The experiments on high-dimensional data are impressive. However, the comparative analysis in terms of time and memory efficiency against the two papers I previously mentioned has still not been conducted; instead, two different papers are referenced. Therefore, I have decided to maintain my current score.

---

> > > ### Author Response · Authors · 2026-04-06
> > >
> > > As suggested, we have now added a direct comparison to latent space methods based on pullback metrics [1,2], and updated the linked document with these results, see Figure 4 and Table 2 in https://anonymous.4open.science/r/icml2026_anonymous-058C/rebuttal_figures.pdf.
> > >
> > >
> > > **Interpolation quality.** We evaluate interpolation in latent space using two VAEs (d=8, 2M parameters and d=32, 12M parameters). We compare (i) linear interpolation, (ii) Riemannian interpolation using metric matching trained directly in latent space, and (iii) Riemannian interpolation using the pullback metric as in [1,2].
> > >
> > > We find that all methods produce comparable interpolants, with only minor differences. In particular, metric matching and pullback metrics yield very similar trajectories in the higher-capacity (d=32) setting, suggesting that they recover closely aligned geometries.
> > >
> > > **Computational cost.**
> > > We additionally compare the cost of constructing the metric. Metric matching is substantially more efficient:
> > > - Small VAE: ~400× faster, ~13× less memory
> > > - Large VAE: >2000× faster, ~13× less memory
> > >
> > > These results directly address the reviewer’s question: while achieving comparable geometric behavior, metric matching avoids the expensive Jacobian computations required by pullback methods.
> > >
> > > We thank the reviewer for this suggestion, which has strengthened the empirical comparison, and hope this resolves the concern regarding latent-space baselines.

---

### Official Review · Reviewer_dVBH · 2026-03-16

**Soundness:** 4
**Presentation:** 3
**Significance:** 4
**Originality:** 4
**Overall Recommendation:** 5
**Confidence:** 3

**Summary:**

This paper provides a new perspective to analyse data manifolds. The authors first discuss the Carré du champ (CDC) identity and its relation to the Riemannian metric. They used the diffusion to derive a simple and tractable form of CDC and train a neural network to predict CDC operator, which can then be used to address various developments, including interpolation, on a data manifold.

**Compliance With Llm Reviewing Policy:**

Affirmed.

**Key Questions For Authors:**

Admittedly, this work is beyond my geometry knowledge (kinda feel good to see the machine matches me to such a beautiful piece of work but I wish I could contribute better technically in my review). That said, I have a couple of questions; hopefully, it helps gain more insights about the method.
1- Have you observed odd eigenvectors for MNIST experiment in Fig.3? If yes, what do they look like or present?
2- Is it possible to study the out-of-distribution behaviour of the metric identification? For example, if on MNIST, the model is trained on 9 digits and evaluated on the 10th, what would happen?
3- For the interpolation, can we see morphing from one digit to another for both linear and manifold cases, similar to Fig.4?

**Limitations:**

This is not discussed. Since the output of the network is the metric, one can imagine that the method struggles in high-dimensional manifolds (although the low-rank solution is discussed)

**Strengths And Weaknesses:**

### Soundness
The work is, to my knowledge, sound and quite advanced. I commend the authors for such a significant work.

### Presentation
The paper is written very well despite the content being quite theoretical. That said, I have some comments to improve the clarity, which I will add at the end of this part.

### Significance
The theory developed in this work to use CDC to understand the geometry of data along with several developments based on that (eg., interpolation on the manifold) advances our understanding and techniques to analyse data manifolds and this is a very significant step forward. address
### Originality:
This work is highly original and advances the theory significantly.

Some minor comments:
- Please define x and B_t in line 138
- For eq.2 and eq.3, if Euclidean counterparts are explained, then for a reader not familiar with hardcore differential geometry, following the paper will be much easier.
- Below eq.3, isn’t it better to show that the evaluation is done at $p \in \mathcal{M}$ explicitly (I am talking about $\Gamma_{\Delta_g})$?

- Above eq.2, it is said that the diffusion is related to CDC identity but it is not obvious by just looking at eq.2

- Around line 177, I did not understand how one can define a conditional CDC. Please provide more details

- Please provide more details about what $\mathcal{L}$ is below eq.3. Again, I think providing an easier conceptual explanation about eq.2-3 (possibly by looking at similar concepts in Euclidean space or simple manifolds) helps readers to follow a lot

- What is $t_b$ in Alg.1?

- What is f in part 3 and around $P_\epsilon$? Please define it. The same applies $P_\epsilon$ as soon after the operator $\mathcal{L}_\epsilon$? is defined.

- In line 173, it is said $\Gamma_\epsilon$ from eq.5 is intractable. It is not clear to me why, I understand memory complexity or numerical issues in high-dimensional spaces but intractability is not clear to me.

- What is the meaning of * in defining $p_Y$?

---

> ### Author Rebuttal · Authors · 2026-03-31
>
> We thank the reviewer for the thorough and thoughtful review. We really appreciate the reviewer’s encouragement and appreciation of our work.  To support the replies, we have attached additional figures in an anonymous link: https://anonymous.4open.science/r/icml2026_anonymous-058C/rebuttal_figures.pdf.
>
> The reviewer’s suggestions for improving and clarifying the exposition are very useful and we will incorporate them all into our final submission. To respond to the reviewer’s questions:
>
> **How to define a conditional CDC?** The equation we give is our definition for the ‘conditional CDC’ (we will change the text to make this clearer). It mirrors the distinction between the marginal and conditional paths in flow-based generative modelling, because it is dependent on x (unlike the ‘marginal CDC’).
>
> **What is $t_b$ in Alg.1?** This is a typo! It was meant to be $\epsilon_b$, i.e. the noise level.
>
> **What is $f$ in part 3?** Here $f$ is a function - we will make this clearer.
>
> **How is the marginal CDC intractible?** We’re using the same notion of tractability from generative modelling. Namely, the hard part is the normalisation constant $d_\epsilon$, which involves an expectation over the entire training dataset. If we want to evaluate this marginal form directly, we need to store all the training data and compute that sum over it to get $d_\epsilon$, which is prohibitive for large data and high dimensions.
>
> **What is the meaning of * in defining $p_Y$?** Here * denotes convolution, i.e. augmenting p with samples from N(0, \epsilon I).
> In response to the reviewers key questions:
>
> **Have you observed odd eigenvectors for MNIST experiment in Fig.3? If yes, what do they look like or present?** The eigenvectors are usually quite interpretable, and we haven’t seen anything unexpected in well-trained models. We ran more experiments on other datasets and they also all seem to make sense: see Figure 1 for our new FFHQ benchmark.
>
> **Is it possible to study the out-of-distribution behaviour of the metric identification?** We are very ineterested to study this phenomenon in future work, since diffusion models typically exhibit this sort of transfer learning, and we would hope to inherit those good properties. If the model architecture is suitable (e.g. a CNN for images), then we think that good transfer learning is likely to occur.
>
> **For the interpolation, can we see morphing from one digit to another?** Yes, we can, although the intermediate images are often less plausible. In future work, we plan to incorporate the denoising technique from [1], where optimization steps are ‘cleaned’ by denoising, which we expect to lead to better interpolation paths in this more challenging scenario.
>
> Finally, we will add further discussion of limitations to the final version. Your question about high-dimensional cases is really interesting, and we expect our methods to deal with them about as well as diffusion models do. To investigate this, we have also added experiments on CIFAR-10 (D = 3072), CelebA (D=12288), and FFHQ (D=196608), which are considerably larger than MNIST (D=728). To evaluate geometric quality in these settings and benchmark against other metrics, we introduce a Tangent Inception Distance metric and use it to benchmark our method, as well as other exisiting baselines. We have detailed these experiments in full in our response to reviewer Bh9d, and show the results in the attached link.
>
> We thank the reviewer again for the helpful suggestions and valuable feedback. We welcome the opportunity to further discuss any remaining concerns.
>
> ______
> [1] Kharitenko et al., "Landing with the score: Riemannian optimization through denoising." ICLR, 2026

---

> > ### Author Rebuttal · Reviewer_dVBH · 2026-04-03
> >
> > I thank the authors for addressing my concerns and also for including new experiments. I confirm that I have also read the comments of other reviewers and the rebuttal. I maintain my score as I believe the work has made a significant step in understanding and analyzing data manifolds.

---

> > > ### Author Response · Authors · 2026-04-08
> > >
> > > We thank the reviewer for their time and encouraging appraisal of our work.

---

### Decision · Program_Chairs · 2026-04-30

**Decision:**

Accept (spotlight)

**Comment:**

There is a general consensus that the paper is very strong, technically sound, and highly original. Although the paper is quite theoretical, the reviewers also found it well written. The main concerns were about missing related work and the need for stronger experiments on larger, high-dimensional datasets. The authors used the discussion phase well: they added substantial new comparisons, larger-scale results, and additional information about time/memory costs, which addressed most of the concerns and made the reviewers more positive.

Overall, all reviewers recommend acceptance, and I strongly support that recommendation. I think this is an excellent paper.